# Regulation of the evolutionarily conserved muscle myofibrillar matrix by cell type dependent and independent mechanisms

Peter T. Ajayi [1,4], Prasanna Katti [1,4], Yingfan Zhang[1], T. Bradley Willingham[1], Ye Sun[2], Christopher K. E. Bleck [2] & Brian Glancy [1,3✉]

Skeletal muscles play a central role in human movement through forces transmitted by contraction of the sarcomere. We recently showed that mammalian sarcomeres are connected through frequent branches forming a singular, mesh-like myofibrillar matrix. However, the extent to which myofibrillar connectivity is evolutionarily conserved as well as mechanisms which regulate the specific architecture of sarcomere branching remain unclear. Here, we demonstrate the presence of a myofibrillar matrix in the tubular, but not indirect flight (IF) muscles within *Drosophila melanogaster*. Moreover, we find that loss of transcription factor *H15* increases sarcomere branching frequency in the tubular jump muscles, and we show that sarcomere branching can be turned on in IF muscles by *salm*-mediated conversion to tubular muscles. Finally, we demonstrate that *neurochondrin* misexpression results in myofibrillar connectivity in IF muscles without conversion to tubular muscles. These data indicate an evolutionarily conserved myofibrillar matrix regulated by both cell-type dependent and independent mechanisms.

[1] Muscle Energetics Laboratory, NHLBI, NIH, Bethesda, MD 20892, USA. [2] Electron Microscopy Core, NHLBI, NIH, Bethesda, MD 20892, USA. [3] NIAMS, NIH, Bethesda, MD 20892, USA. [4] These authors contributed equally: Peter T. Ajayi, Prasanna Katti. ✉email: Brian.glancy@nih.gov

Sarcomeres are the basic force production unit of the striated muscle cell and make up the bulk of the muscle volume[1,2]. Not surprisingly, improper sarcomere assembly, maintenance, and contractile function have been linked to a wide variety of pathologies including dystrophy, aging, and heart failure[3–7]. We recently demonstrated that mammalian sarcomeres are not arranged end-to-end into many individual, parallel myofibrils as had long been thought[8–10], but instead, sarcomeres frequently branch to create a mesh-like myofibrillar matrix thereby providing a direct pathway for active force transmission across both the length and width of the muscle cell[11]. Sarcomere branching frequency has been shown to vary according to cell type[11,12], though the mechanisms regulating the degree of connectivity have yet to be determined. Additionally, evidence of sarcomere branching can be seen in muscle images from fish[13], chickens[14,15], and frogs[16] in addition to mice[12,17] and humans[11,18,19] suggesting that myofibrillar networks may be conserved across vertebrate species. Conversely, insect fibrillar muscles, such as the commonly studied *Drosophila* indirect flight (IF) muscle, are known to be made up of the textbook individual myofibrils running the entire length of the muscle cell[20–22], implying that a myofibrillar matrix may not occur in invertebrate muscles. However, the majority of *Drosophila* muscle cells are tubular, rather than fibrillar in nature[23], and the connectivity of sarcomeres within tubular muscles remains unclear.

Here we use high-resolution, 3D electron microscopy to determine to what extent myofibrillar connectivity occurs across *Drosophila melanogaster* muscles. We hypothesized that *Drosophila* tubular muscles form myofibrillar networks which would open this powerful genetically tractable model system[24] for studies into the mechanisms of sarcomere branching. Indeed, we show that the myofibrillar matrix is conserved in the invertebrate tubular muscles and that the frequency of sarcomere branching in tubular muscles is cell-type dependent with greater myofibrillar connectivity in the relatively more oxidative leg and direct flight (DF) muscles than in the glycolytic jump (tergal depressor of the trochanter, TDT) muscles. Further, we demonstrate that the muscle-specific loss of cell-type specification factor *H15*[25] leads to the conversion of the jump muscles to a leg muscle-like phenotype with increased sarcomere branching and smaller myofibrils while maintaining the tubular nature of the contractile apparatus. The complete lack of sarcomere branching in the fibrillar IF muscles allowed for the demonstration of two types of regulatory on/off switches for myofibrillar connectivity. First, loss of IF muscle specification factor *salm*[22] leads to conversion of the IF muscles to a tubular phenotype, resulting in sarcomere branching linking the contractile apparatus across the width of the cell. Second, we identify *neurochondrin* (*NCDN*) as a cell-type independent regulator of sarcomere branching in the IF muscles. Misexpression of *NCDN* during myofibril assembly leads to the formation of thin myofibrillar branches that link the larger fibrillar segments of the IF contractile apparatus together, resulting in the formation of a mesh-like myofibrillar matrix without conversion to a tubular phenotype. Thus, we find that connectivity of the evolutionarily conserved myofibrillar matrix can be regulated by both cell-type dependent (*salm*, *H15*) and cell-type independent (*NCDN*) mechanisms.

## Results

### Myofibrillar matrix is conserved in Drosophila tubular muscle.
To investigate myofibrillar connectivity in the invertebrate striated muscles of *D. melanogaster*, we used focused ion beam scanning electron microscopy (FIB-SEM) to visualize muscle cell ultrastructure with 10 nm pixels along all three dimensions (3D)[11,26]. The 26 IF muscles (12 dorsal longitudinal + 14 dorsoventral muscles) are the largest muscle group in *Drosophila* and are known as fibrillar muscles due to the individual fiber-like nature of their myofibrils which can be observed under a light microscope[20,22,23]. Not surprisingly, our FIB-SEM analyses of the fibrillar dorsal longitudinal muscles ($n = 3$ cells, 114 myofibrils, and 982 sarcomeres) corroborates the individual nature of the circular myofibrils running the length of the IF muscle cell without any connecting sarcomere branches (Fig. 1a, e, f). On the other hand, *Drosophila* tubular muscles have centralized nuclei surrounded by cross-striated myofibrils in tube-like arrays that make up the majority of the adult muscle cells[23]. To evaluate tubular muscles with varying developmental backgrounds and functional demands[27–30], we chose to assess three different tubular muscle types: DF, TDT, and leg muscles. While the tightly packed nature of the tubular muscle contractile apparatus has prohibited analyses of myofibrillar connectivity using standard light microscopy approaches (>200 nm resolution), we are able to clearly delineate the sarcomere boundaries in tubular muscles due to the high resolution of FIB-SEM as well as the relatively higher content of the sarcotubular system (sarcoplasmic reticulum + t-tubules, SRT) which runs in between each sarcomere (Supplementary Fig. 1). Tracking the myofibrillar structures along the length of the tubular muscles revealed that sarcomere branching does indeed occur in invertebrate muscles, as myofibrillar networks are found in the *Drosophila* DF, TDT, and leg muscles (Fig. 1b–d). Moreover, the amount of sarcomere branching varied among the three tubular muscle types despite all three having the same primary actin 79B isoform[31]. The percentage of myofibrils within the field of view of our datasets (up to $50 \times 30 \times 27\,\mu m$) with at least one branching sarcomere (Fig. 1e), as well as the percentage of branching sarcomeres (Fig. 1f) was highest in the leg muscles followed by the DF and TDT muscles ($94 \pm 3.5\%$ (mean ± SE) of myofibrils, $65.9 \pm 3.4\%$ of sarcomeres, $n = 3$ cells, 150 myofibrils, 690 sarcomeres in leg muscles (Supplementary Video 1); $77 \pm 4.4\%$ of myofibrils, $32.4 \pm 5.0\%$ of sarcomeres, $n = 4$ cells, 200 myofibrils, 1461 sarcomeres in DF muscles; and $50 \pm 3.1\%$ of myofibrils, $10.8 \pm 3.1\%$ of sarcomeres, $n = 3$ cells, 150 myofibrils, 1713 sarcomeres in jump muscles (Supplementary Video 2)). Thus, the myofibrillar matrix is evolutionarily conserved in invertebrate muscles, and like mammalian muscle[11], the connectivity of the myofibrillar matrix is cell-type dependent.

Also similar to mammalian muscle[11], sarcomere branching in *Drosophila* muscle can occur through different physical mechanisms. The simplest type of sarcomere branch is termed a single branching event where one sarcomere segment transitions into two segments (Supplementary Fig. 2b, d and Supplementary Video 3). More complex branching types can be grouped as multi-branching events and occur when a single sarcomere segment transitions to three or more resultant segments (Supplementary Fig. 2c, e and Supplementary Video 4). Thus, the total number of sarcomere branches can be greater than the number of sarcomeres, such as in the leg muscle where $38.7 \pm 2.9\%$ of sarcomeres undergo a multi-branching event (Supplementary Fig. 2e) and $27.2 \pm 1.2\%$ of sarcomeres undergo a single branching event (Supplementary Fig. 2d) resulting in $13.0 \pm 1.2$ sarcomere branches per 10 sarcomere lengths (Supplementary Fig. 2f). Further, to determine whether sarcomere branching was related to subcellular location, we assessed the positions of all sarcomere branches in the highly branching leg muscle and the low branching TDT muscle relative to the cell boundary (Supplementary Fig. 3). In both the leg and TDT muscles, sarcomere branches were more likely to occur near the cell periphery than in the interior (Supplementary Fig. 3c, d) similar to previous findings in fish muscle[13].

To evaluate how myofibrillar connectivity may be related to muscle function in *Drosophila*, we assessed the overall cellular content of the contractile apparatus, mitochondria, and the SRT

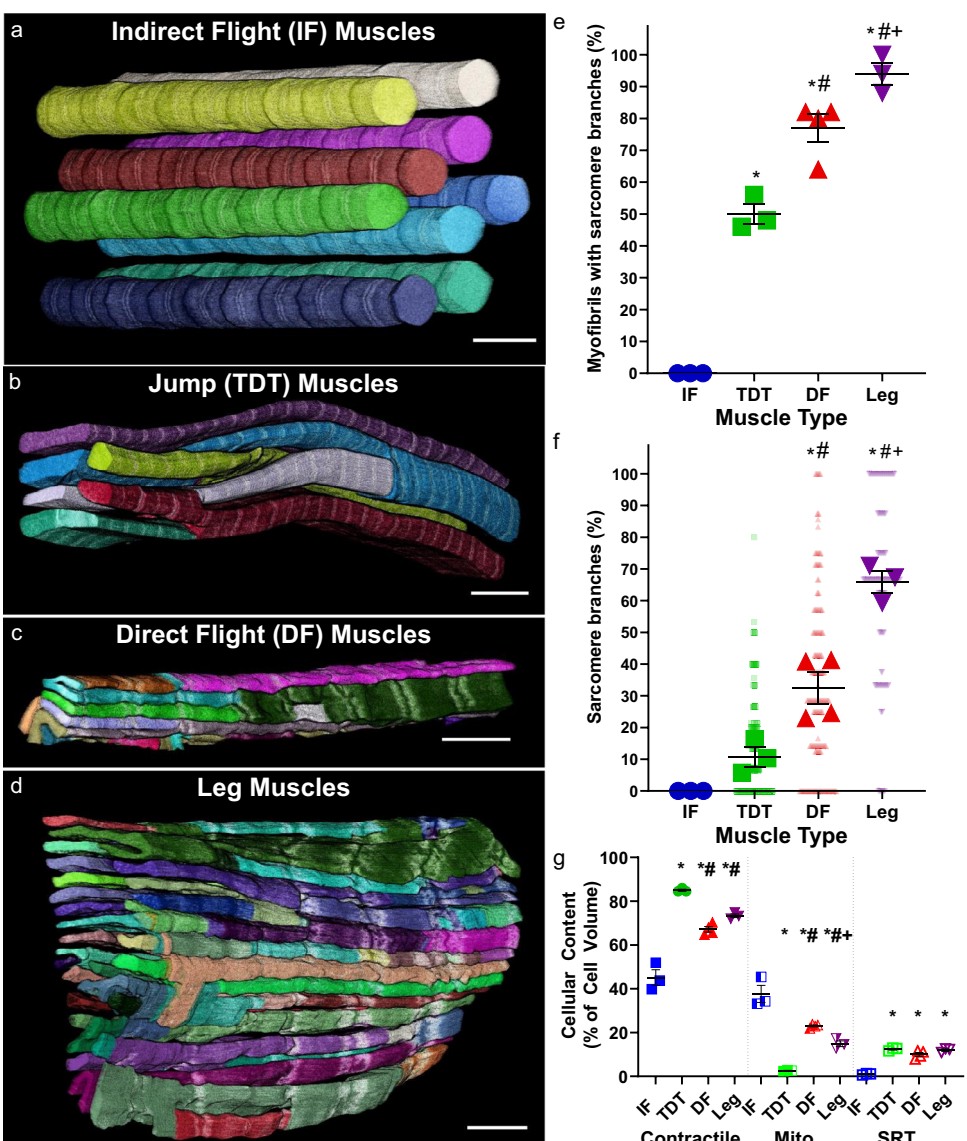

**Fig. 1 Myofibrillar networks in *Drosophila* skeletal muscles. a** 3D rendering of isolated myofibrils in fibrillar indirect flight (IF) muscles. **b–d** 3D rendering of unified myofibrillar matrix structures in tubular muscles. **b** Jump (tergal depressor of the trochanter, TDT) muscles; **c**—direct flight (DF) muscles; **d**—leg muscles. Individual colors represent different myofibrillar segments separated by sarcomere branches. **e** Percentage of myofibrils with at least one branching sarcomere in wild-type IF, TDT, DF, and leg muscles. Asterisk (*): Significantly different from fibrillar indirect flight (one-way ANOVA, $P < 0.0001$). Hash sign (#): Significantly different from tubular TDT (DF: $P = 0.0015$, Leg: $P < 0.0001$). Plus sign (+): Significantly different from tubular DF ($P = 0.0267$). **f** Percentage of sarcomeres per myofibril with at least one branch in wild-type IF, TDT, DF, and leg muscles. Asterisk (*): Significantly different from fibrillar indirect flight (DF: $P = 0.0008$, Leg: $P < 0.0001$). Hash sign (#): Significantly different from tubular TDT (DF: $P = 0.0117$, Leg: $P < 0.0001$). Plus sign (+): Significantly different from tubular DF ($P = 0.0006$). **g** Total percentage of cellular volume occupied by contractile, mitochondrial, and sarcotubular structures in wild-type IF, TDT, and leg muscles. Asterisk (*): Significantly different from fibrillar indirect flight (Contractile—All: $P < 0.0001$; Mito—DF: $P = 0.0015$; TDT & Leg: $P < 0.0001$). Hash sign (#): Significantly different from tubular TDT (DF: Contractile—$P = 0.0003$, Mito—$P = 0.0001$; Leg: Contractile—$P = 0.0092$, Mito—$P = 0.0072$). Plus sign (+): Significantly different from tubular DF (Mito—$P = 0.0495$). $N$ values for **e–g**: IF—3 muscle cells, 3 datasets, 114 myofibrils, 982 sarcomeres; TDT—3 muscle cells, 2 datasets, 160 myofibrils, 1863 sarcomeres; DF—4 muscle cells, 1 dataset, 200 myofibrils, 1461 sarcomeres; and leg—3 muscle cells, 2 datasets, 150 myofibrils, 690 sarcomeres. Larger shape symbols represent data from a single cell and smaller shape symbols represent data from a single myofibril. Bars represent muscle cell overall mean ± SE. Scale bars—2 μm.

system as estimates of the capacity for force production[32], oxidative phosphorylation[33], and calcium cycling[33], respectively. TDT muscles have the highest contractile content ($85.0 ± 0.5\%$ of cell volume, $n = 3$ cells) and the lowest mitochondrial content ($2.6 ± 0.1\%$, $n = 3$), whereas the IF muscles have the lowest contractile content ($45.1 ± 3.6\%$, $n = 3$) and highest mitochondrial content ($37.6 ± 3.9\%$, $n = 3$) (Fig. 1g). However, the less frequent sarcomere branching in the IF and TDT muscles compared to the DF and leg muscles (Fig. 1f) suggests that the

degree of sarcomere branching is not directly related to myofibrillar or mitochondrial content, and thus the capacities for force production or oxidative energy conversion, respectively. Conversely, the non-branching IF muscles have much lower SRT content ($1.0 ± 0.2\%$, $n = 3$) than each of the tubular muscles (TDT: $12.4 ± 0.4\%$, $n = 3$; DF: $10.0 ± 0.9\%$, $n = 4$; leg: $11.9 ± 0.5\%$, $n = 3$) (Fig. 1g) implying that myofibrillar connectivity may be related to SRT content and perhaps calcium cycling. Indeed, IF muscles are asynchronous in nature[34] meaning that a single cycle

of calcium can result in tens of muscle contractions, whereas the tubular muscles, like their mammalian counterparts, are synchronous and require calcium to cycle for each contraction. However, no differences in SRT content are observed across the three tubular muscles here despite the more than six-fold variation in sarcomere branching percentage (Fig. 1f). Thus, these data suggest that while SRT content and perhaps calcium cycling may be related to the need for initiation of myofibrillar networks, there is no apparent relationship between SRT content and the degree of connectivity within myofibrillar networks.

**H15 regulates sarcomere branching in tubular TDT muscles.** We recently identified T-box transcription factor *H15* as capable of specifying between the TDT and leg muscle phenotypes based on conversion of the mitochondrial network configuration in muscle-specific *H15* knockdown (KD) tubular muscles (*Mef2-Gal4; UAS-H15 RNAi*) as observed with confocal microscopy[25]. Based on the large difference in the magnitude of sarcomere branching between the TDT and leg muscles shown here (Fig. 1), we hypothesized that *H15* KD would also regulate myofibrillar network connectivity through conversion of the TDT muscles to a leg muscle phenotype. *H15* KD in the TDT muscles resulted in a reduction in cross-sectional area (CSA) and circularity per myofibril ($0.73 \pm 0.15 \ \mu m^2$, $0.48 \pm 0.05$) compared to the wild-type TDT muscles ($0.97 \pm 0.02 \ \mu m^2$, $0.53 \pm 0.01$) and closer to the level of the leg muscles ($0.56 \pm 0.15 \ \mu m^2$, $0.46 \pm 0.02$) (Fig. 2c). Moreover, loss of *H15* led to the formation of holes in the TDT muscle z-disks (Supplementary Videos 5 and 6). Importantly, *H15* KD in the TDT raised the percentage of myofibrils with at least one sarcomere branch ($95.0 \pm 5.0\%$, $n = 2$ cells) to levels similar to those observed in the wild type leg ($94.0 \pm 3.5\%$, $n = 3$) (Fig. 2b, d and Supplementary Video 7). The percentage of sarcomeres with a branch in the *H15* KD TDT muscles ($78.2 \pm 2.3\%$) increased 7.2-fold over the wild-type TDT muscles ($10.8 \pm 3.1\%$) and was also no different from the wild-type leg ($65.9 \pm 3.4\%$) (Fig. 2d). Additionally, the frequency of sarcomere branches was also similar between the *H15* KD TDT ($14.0 \pm 0.4$ branches per 10 sarcomere lengths) and leg muscles ($13.0 \pm 1.2$ branches per 10 sarcomere lengths) and higher than in the wild type TDT muscles ($1.8 \pm 0.5$ branches per 10 sarcomere lengths) (Fig. 2e). Overall, these data suggest that *H15* regulates the connectivity of myofibrillar networks in the tubular TDT muscle by preventing these muscles from taking a leg muscle phenotype.

**Loss of *salm* initiates tubular IF muscle myofibrillar networks.** Due to the absence of sarcomere branching in the IF muscles and the presence of myofibrillar networks in each of the three tubular muscles assessed above (Fig. 1), we hypothesized that factors which control the fibrillar/tubular fate of IF muscles such as *extradenticle* (*exd*)[35], *homothorax* (*hth*)[35], *vestigial* (*vg*)[22], *ladybird early* (*lbe*)[22], *salm*[22], and *H15*[25] would also serve effectively as a switch for myofibrillar connectivity in IF muscles. To test this hypothesis, we evaluated sarcomere branching in the IF muscles of muscle-specific *salm* KD (*Mef2-Gal4; UAS-salm RNAi*) flies which are known to convert to a tubular muscle phenotype[22,36]. 3D rendering of the *salm* KD contractile apparatus confirmed that knocking down *salm* in IF muscles changes its organization to resemble the cross-striated nature of tubular muscles and initiated myofibrillar connectivity (Fig. 3b and Supplementary Video 8). Consistently, loss of *salm* leads to a reduction in CSA ($0.86 \pm 0.11 \ \mu m^2$) and circularity ($0.66 \pm 0.06$) per myofibril compared to the wild type IF muscles ($2.15 \pm 0.03 \ \mu m^2$, $0.88 \pm 0.00$) (Fig. 3c). Moreover, similar to the loss of *H15* in the TDT muscle, *salm* KD results in the formation of holes in the z-disks of the IF muscle (Supplementary Video 9). The high degree

of myofibrillar connectivity across the length and width of the *salm* KD muscle is apparent in the 3D rendering by the frequent changes in color representing the location of sarcomere branches (Fig. 3b). Quantitative assessment of myofibrils within the field of view of our datasets revealed the extent sarcomeres connect with one another, and its frequency was comparable to that of tubular muscles we evaluated. Assessing 3 cells, 180 myofibrils, and 932 sarcomeres, $85.3 \pm 10.1\%$ of myofibrils had at least one sarcomere that branched to connect with adjacent sarcomeres and $45.2 \pm 17.5\%$ of sarcomeres were branched (Fig. 3d), yielding a total of $6.9 \pm 3.8$ sarcomere branches per 10 sarcomere lengths when accounting for both single and multi-branching events (Fig. 3e). Thus, the role of *salm* as a master regulator of IF muscle cell type[22,36] also involves the inhibition of myofibrillar network formation.

**NCDN regulates sarcomere branching within fibrillar IF muscles.** We identified *NCDN* as a gene potentially involved in sarcomere branching by filtering our recent proteomic database of *Drosophila* IF, TDT, leg, and *salm* KD IF muscles[25] for proteins that are upregulated in muscles proportional to the sarcomere branching frequencies described above (leg > TDT > IF and *salm* KD > IF). NCDN is a cytoplasmic, leucine-rich protein conserved in humans and primarily expressed in the brain with a role in neurite outgrowth through modulation of metabotropic glutamate receptor 5 signaling, $Ca^{2+}$/calmodulin-dependent protein kinase II phosphorylation, and actin dynamics[37–40]. Little is known about the role of *NCDN* in striated muscle, though its expression is upregulated in a *Drosophila* model of hypercontraction-induced myopathy[41]. To investigate the role of *NCDN* in myofibrillar network connectivity, we generated muscle-specific *NCDN* KD (*Mef2-Gal4; UAS-NCDN RNAi*) and overexpression (OE, *Mef2-Gal4; UAS-NCDN*) flies. Both *NCDN* misexpression fly lines are viable but display either a complete loss (*NCDN* KD) or weak (*NCDN* OE) flight ability, indicating a critical role for *NCDN* in muscle function. *NCDN* KD was confirmed by the more than ten-fold decrease in *NCDN* mRNA expression in the KD muscles as measured by qPCR (Supplementary Fig. 4g), whereas *NCDN* OE results in a smaller, 57.2% increase in *NCDN* mRNA expression (Supplementary Fig. 4h). Additionally, *NCDN* KD does not alter the protein expression of fibrillar muscle specification factors Salm[22] and H15[25] compared to wild-type IF muscles nor alter the configuration of mitochondrial networks running in parallel to the myofibrils (Supplementary Fig. 5, Supplementary Table 1). We initially screened the *NCDN* KD and OE IF muscles under the light microscope by visualizing sarcomeric actin stained with phalloidin (Supplementary Fig. 4a–f). While the *NCDN* KD and OE muscles appear to largely maintain their individual fibrillar nature, sarcomere branching can be seen in the longitudinal images, and many thin actin projections are observed in the cross-sectional images (white arrows, Supplementary Fig. 4a–f).

Since both *NCDN* KD and OE result in sarcomere branching under the light microscope (Supplementary Fig. 4), the stronger phenotype of *NDCN* KD muscles was chosen for the more detailed FIB-SEM analyses to better visualize and quantify myofibrillar connectivity in the *NCDN* misexpressed IF muscles. The majority of the contractile volume in *NCDN* KD IF muscle is comprised of large, roughly circular sarcomeres in series (Fig. 4b) similar to the wild-type IF muscle (Fig. 4a). However, thin fiber-like myofibrillar segments containing as few as four myosin filaments project out from the larger segments and unite them together (Fig. 4e–f and Supplementary Videos 10 and 11). As a result, the average CSA per myofibril is smaller in the *NCDN* KD IF muscles ($1.14 \pm 0.32 \ \mu m^2$) compared to wild type IF muscles ($2.15 \pm 0.03 \ \mu m^2$) though some myofibrils are much larger ($6 + \mu m^2$) in the *NCDN* KD muscle

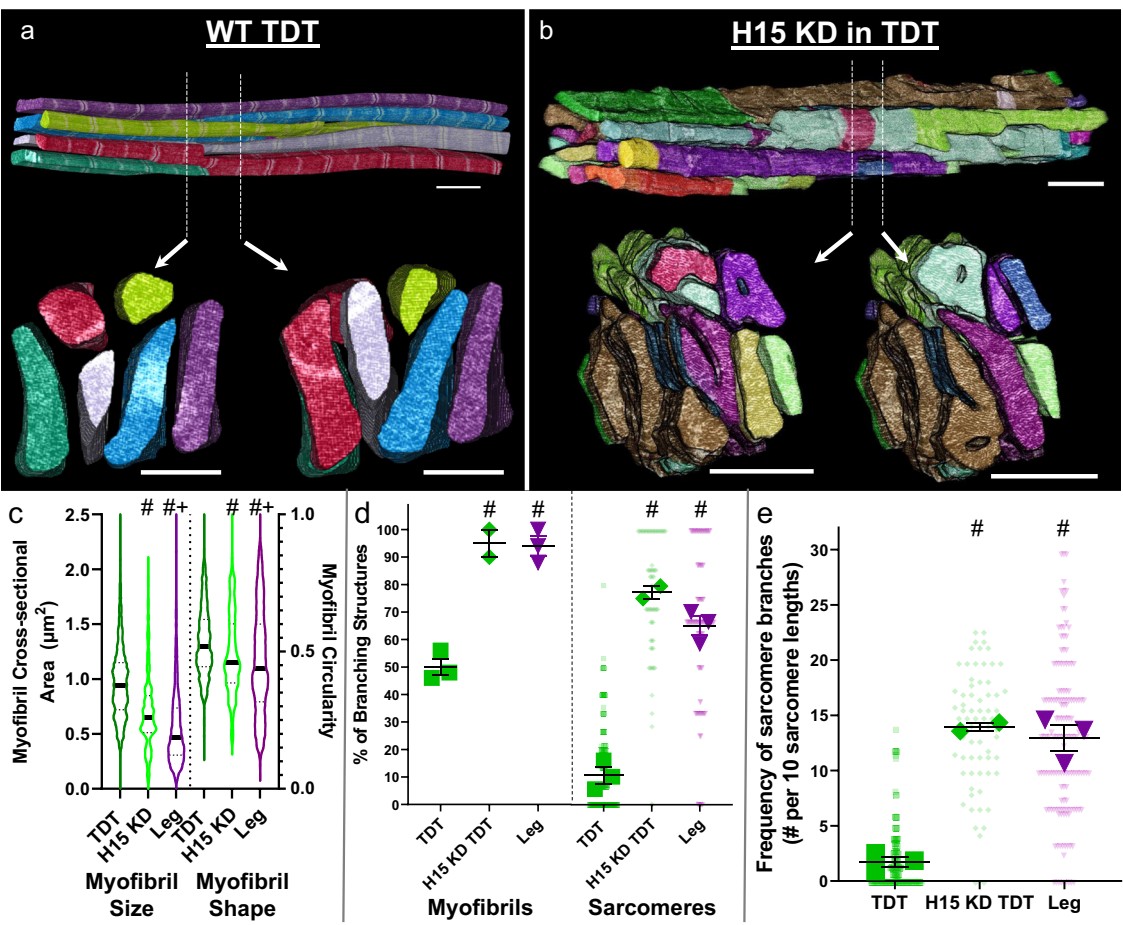

**Fig. 2 Loss of *H15* increases TDT muscle sarcomere branching to leg muscle levels. a, b** 3D rendering of myofibrillar matrix in TDT muscles with (**b**) and without (**a**) Mef2-Gal4 driven RNAi knockdown of *H15*. Bottom two images show clippings through the long axis of the muscle directly above. **c** Assessment of myofibril size and shape for TDT, *H15* KD TDT, and leg muscles. **d** Percentage of myofibrils with at least one branching sarcomere (left) and percentage of sarcomeres per myofibril with a branch (right). **e** Frequency of sarcomere branching in TDT, *H15* KD TDT, and leg muscles. *N* values: TDT—3 muscle cells, 2 datasets, 160 myofibrils, 1863 sarcomeres; *H15* KD TDT—2 muscle cells, 1 dataset, 70 myofibrils, 403 sarcomeres; and leg—3 muscle cells, 2 datasets, 150 myofibrils, 690 sarcomeres. Larger shape symbols represent data from a single cell and smaller shape symbols represent data from a single myofibril. Bars represent muscle cell overall mean ± SE. Hash sign (#): Significantly different from tubular TDT (*P* < 0.01). Plus sign (+): Significantly different from H15 KD TDT (one-way ANOVA, *P* < 0.05). Scale bars—2 μm.

(Fig. 4g). Conversely, taking into account the relative volume of each myofibril by measuring the volume-weighted average shows that the average myofibril CSA is larger for *NCDN* KD (2.86 μm$^2$) than in the wild type (2.16 μm$^2$, gray boxes, Fig. 4g). The circularity of the *NCDN* KD IF muscle myofibrils (0.74 ± 0.02) is higher than in any of the tubular muscles assessed here but not as high as the nearly perfect circular profiles of the wild type IF muscles (0.88 ± 0.00) (Fig. 4g). Additionally, unlike the *H15* KD TDT and *salm* KD IF muscles, the formation of holes in the z-disks in the *NCDN* KD IF muscles is rare (Supplementary Video 12). Instead, many z-disks retain a circular profile similar to the wild-type IF muscle, while others appear stretched or pulled along the longitudinal axis of the muscle often forming thin fiber-like strands several microns in length connecting to adjacent *z*-disks (Supplementary Video 13) and reminiscent of *z*-disk streaming[42,43]. To quantify the connectivity of myofibrillar networks in the *NCDN* KD IF muscles, we assessed the structures of 100 myofibrils and 878 sarcomeres across three different cells. Every single myofibril had at least one sarcomere branch and 53.8 ± 7.9% of sarcomeres branched in the *NCDN* KD IF muscles (Fig. 4h). Overall, these data demonstrate that *NCDN* acts as a regulator of myofibrillar network connectivity without altering the fibrillar fate of IF muscles.

To better understand the role of *NCDN* in regulating sarcomere branching, we performed *NCDN* KD at different windows of time during IF muscle maturation to determine the developmental period(s) in which *NCDN* expression is critical for myofibrillar network formation. Loss of *NCDN* from early myoblast formation through ~24 h APF (*1151-Gal4; UAS-Dicer2*[44]) resulted in a decrease in flight ability (Fig. 5v) but did not lead to the formation of branching myofibrillar networks in the adult IF muscles (Fig. 5c). Similarly, loss of *NCDN* from ~48 h APF through adulthood (*Mhc-Gal4*[24,45]) reduced flight ability (Fig. 5v) and also did not result in the formation of myofibrillar networks in adult IF muscles (Fig. 5e). Conversely, *NCDN* KD from ~24 h APF through adulthood (*Act88F-Gal4*[46]) led to a complete loss of flight ability (Fig. 5v) and resulted in myofibrillar networks connected by branching sarcomeres (Fig. 5d) in adult IF muscles. These data suggest that *NCDN* regulation of branching myofibrillar network formation in IF muscles occur between 24 and 48 h APF. Consistently, developmental timecourse analyses of pupal IF muscles lacking *NCDN* from early myoblast formation through the lifespan (*Mef2-Gal4*[47]) reveal that loss of *NCDN* leads to myofibrillar defects beginning between 32 and 48 h APF (Fig. 5f–u). Moreover, previous RNAseq analyses of

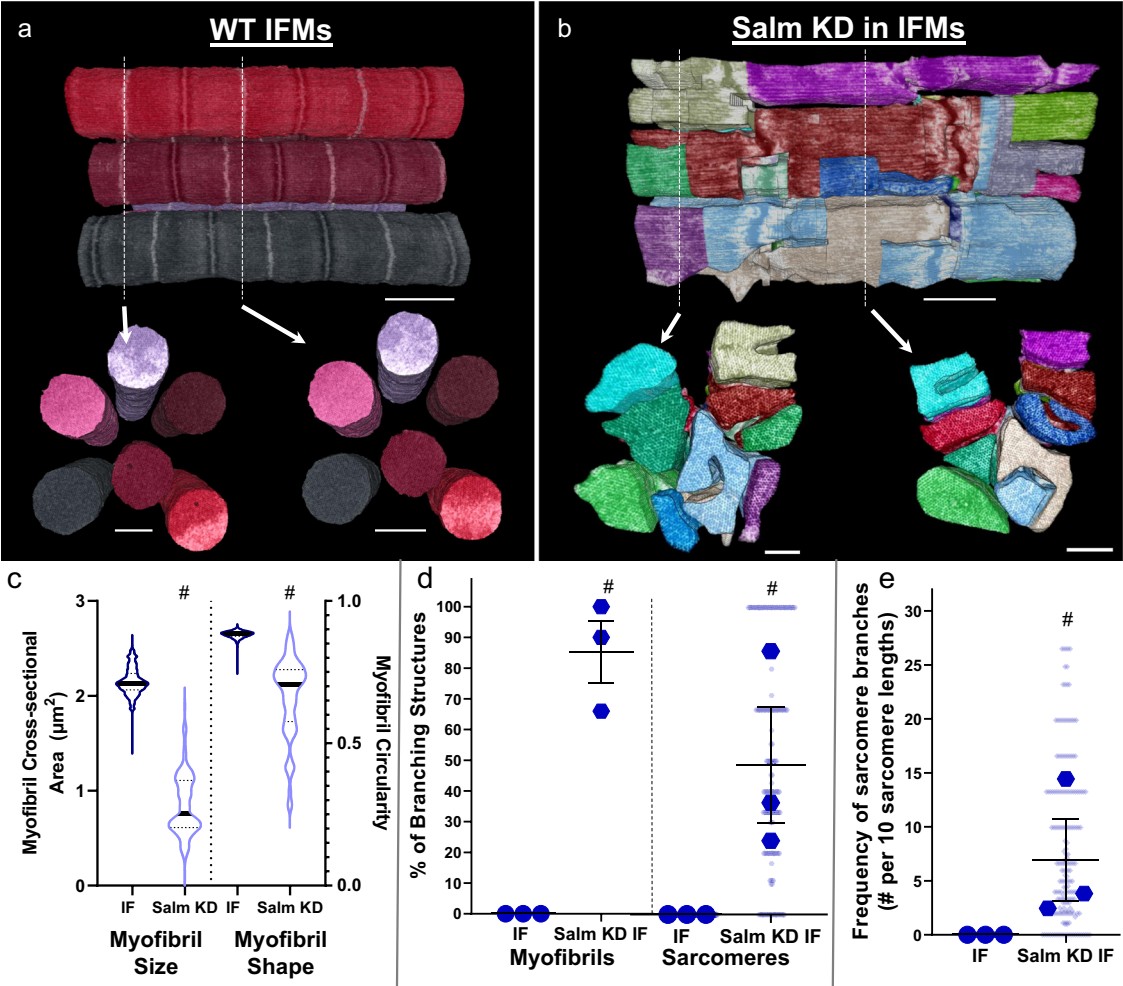

**Fig. 3 Loss of *salm* initiates sarcomere branching through tubular conversion of indirect flight muscles. a** 3D rendering of myofibrillar matrix in IF muscles with (**b**) and without (**a**) Mef2-Gal4 driven RNAi knockdown of *salm*. Bottom two images show clippings through the long axis of the muscle directly above. **c** Assessment of myofibril size and shape for IF and *Salm* KD IF muscles. **d** Percentage of myofibrils with at least one branching sarcomere (left) and percentage of sarcomeres per myofibril with a branch (right). **e** Frequency of sarcomere branching in IF and *Salm* KD IF muscles. *N* values:: IF—3 muscle cells, 3 datasets, 114 myofibrils, 982 sarcomeres; *Salm* KD IF—3 muscle cells, 3 datasets, 180 myofibrils, 932 sarcomeres. Larger shape symbols represent data from a single cell and smaller shape symbols represent data from a single myofibril. Bars represent muscle cell overall mean ± SE. Hash sign (#): Significantly different from fibrillar IF (two-sided, independent t-test, $P < 0.05$). Scale bars—1 μm.

flight muscle transcript abundance during pupal development[48] demonstrated that *NCDN* expression starts to increase at 16 h APF, peaks sharply at 30 h APF, and stabilizes at low expression levels from 72 h APF to adulthood. Overall, these data indicate that *NCDN* regulation of myofibrillar network formation occurs during the myofibril assembly stage of IF muscle development[24] after myoblasts have already fused and myotubes have attached to tendons.

Since both *NCDN* KD and *NCDN* OE muscles show branching sarcomeres, we hypothesized that *NCDN* serves to balance the myofibril assembly process[49,50]. Indeed, altering the balance between actin and myosin expression in IF muscles through the expression of heterozygous null alleles for either IF muscle actin (Act88F) or myosin heavy chain (Mhc) has been previously shown to lead to sarcomere branches connecting adjacent myofibrils in IF muscles[49]. Conversely, IF muscles with a balanced reduction of both actin and myosin largely maintain their individual myofibril structures[49]. To test whether *NCDN* may regulate the actin-myosin balance in IF muscles, we performed qPCR to assess Act88F and Mhc RNA expression in adult flies with and without *NCDN* KD at different developmental stages.

*NCDN* KD at any stage does not alter Mhc expression levels compared to controls (Fig. 5w, x). Consistently, *NCDN* KD (*Mef2-Gal4*) IF muscles expressing GFP tagged Mhc using the weeP26 gene trap[51] show two GFP spots per sarcomere (Supplementary Fig. 4k–s) similar to controls (Supplementary Fig. 4i–k) and unlike the *salm* KD IF muscles (Supplementary Fig. 4o–q) indicating that loss of *NCDN* does not convert IF muscle Mhc to the tubular muscle Mhc variant. Conversely, *NCDN* KD driven by *Mef2-Gal4* (all stages) and *Act88F-Gal4* (myofibril assembly and maturation stages), but not *Mhc-Gal4* (myofibril maturation stage) led to a reduction in Act88F expression (Fig. 5w, x) compared to controls. Additionally, we evaluated whether the *NCDN* KD mediated loss of the IF muscle actin isoform, Act88F, was compensated for by an increase in the tubular muscle actin isoform, Act79B. Instead, *Mef2-Gal4* driven *NCDN* KD muscles show a reduction in Act79B RNA expression as well as a decrease in two other actin cytoskeleton transcripts, troponin C isoform 4 (TpnC4) and α−actinin (Actn) (Supplementary Fig. 4i, j). On the other hand, *NCDN* KD does not alter the expression of actin polymerization and depolymerization factors, diaphanous (dia) and twinstar (tsr), respectively. Overall, these results show that the

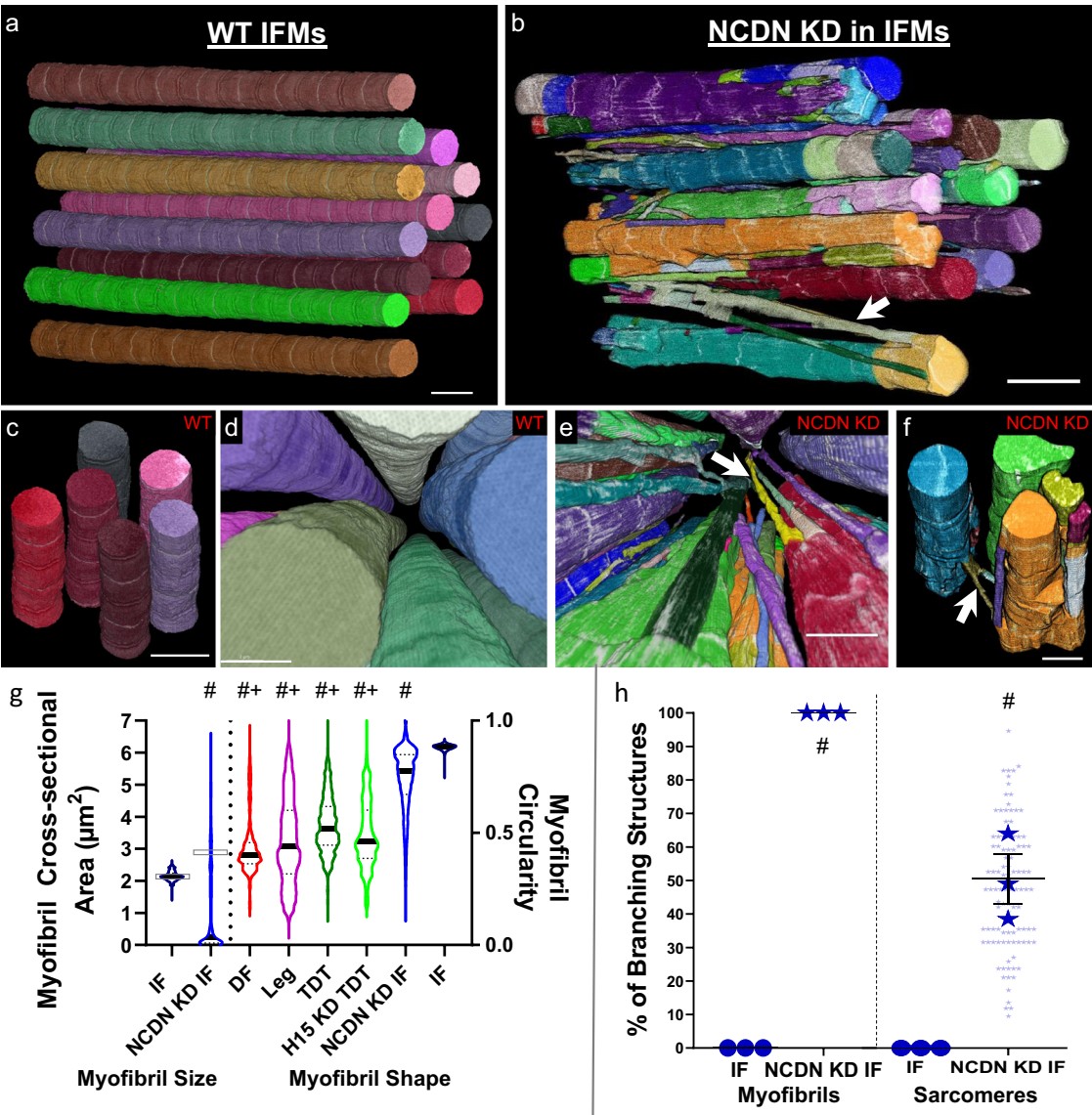

**Fig. 4 Loss of *NCDN* turns on myofibrillar connectivity in fibrillar indirect flight muscles. a–f** 3D renderings of *NCDN* knockdown (**b**, **e**, **f**) and wild-type (**a**, **c**, **d**) IF muscles showing myofibrillar networks at different angles. **g** Assessment of myofibril size per myofibril for IF and *NCDN* KD IF muscles (left) and myofibril shape for tubular muscles, IF, and *NCDN* KD IF muscles (right). Hash sign (#): Significantly different from fibrillar IF. Plus sign (+): Significantly different from *NCDN* KD IF (one-way ANOVA, $P < 0.05$). Gray boxes indicate volume-weighted average. **h** Percentage of myofibrils with at least one branching sarcomere (left) and percentage of sarcomeres that branch (right) in wild type IF and *NCDN* KD IF muscles. N values: IF—3 muscle cells, 3 datasets, 114 myofibrils, 982 sarcomeres and *NCDN* KD IF—3 muscle cells, 2 datasets, 100 myofibrils, 878 sarcomeres. Larger shape symbols represent data from a single cell and smaller shape symbols represent data from a single myofibril. Bars represent muscle cell overall mean ± SE. Hash sign (#): Significantly different from fibrillar IF (two-sided, independent *t* test, $P < 0.01$). Scale bars—2 µm.

loss of *NCDN* during myofibril assembly leading to the formation of branching sarcomeres (Fig. 5a–e) occurs together with a loss of actin-myosin balance resulting from a decrease in actin expression (Fig. 5w, x).

## Discussion

Here we show that the highly connected myofibrillar networks linked together across the width of the muscle cell by branching sarcomeres recently described in mammalian muscles[11] are conserved in the tubular muscles of the invertebrate *D. melanogaster*. In both mammalian skeletal muscles[11] and *Drosophila* tubular muscles, large variability in the degree of connectivity exists across muscle types (Fig. 1), where the relatively more oxidative and lower force-producing muscles (i.e., mammalian slow-twitch and fly DF and leg muscles) have sarcomeres which

branch 3–6 times more frequently than in the high force producing, glycolytic muscles (i.e., mammalian fast-twitch and fly TDT muscles). These data combined with the similarities in the cellular volume occupied by the contractile apparatus, mitochondria, and SRT between the *Drosophila* leg (Fig. 1g) and mammalian slow-twitch oxidative muscles[1,26] and between the TDT (Fig. 1g) and mammalian fast-twitch glycolytic muscles[6,16] suggest that *Drosophila* tubular muscles provide an ideal model system to investigate the mechanisms of myofibrillar connectivity. Conversely, the near-perfect circular cylindrical shape and complete lack of sarcomere branching in the fibrillar IF muscle myofibrils (Fig. 1a, e), in addition to the very low SRT content (Fig. 1g) are unlike any known mammalian muscle, including the often compared cardiac muscle cell[21,22,36,48,52] in which myofibrillar networks are indeed formed[11]. However, we demonstrate

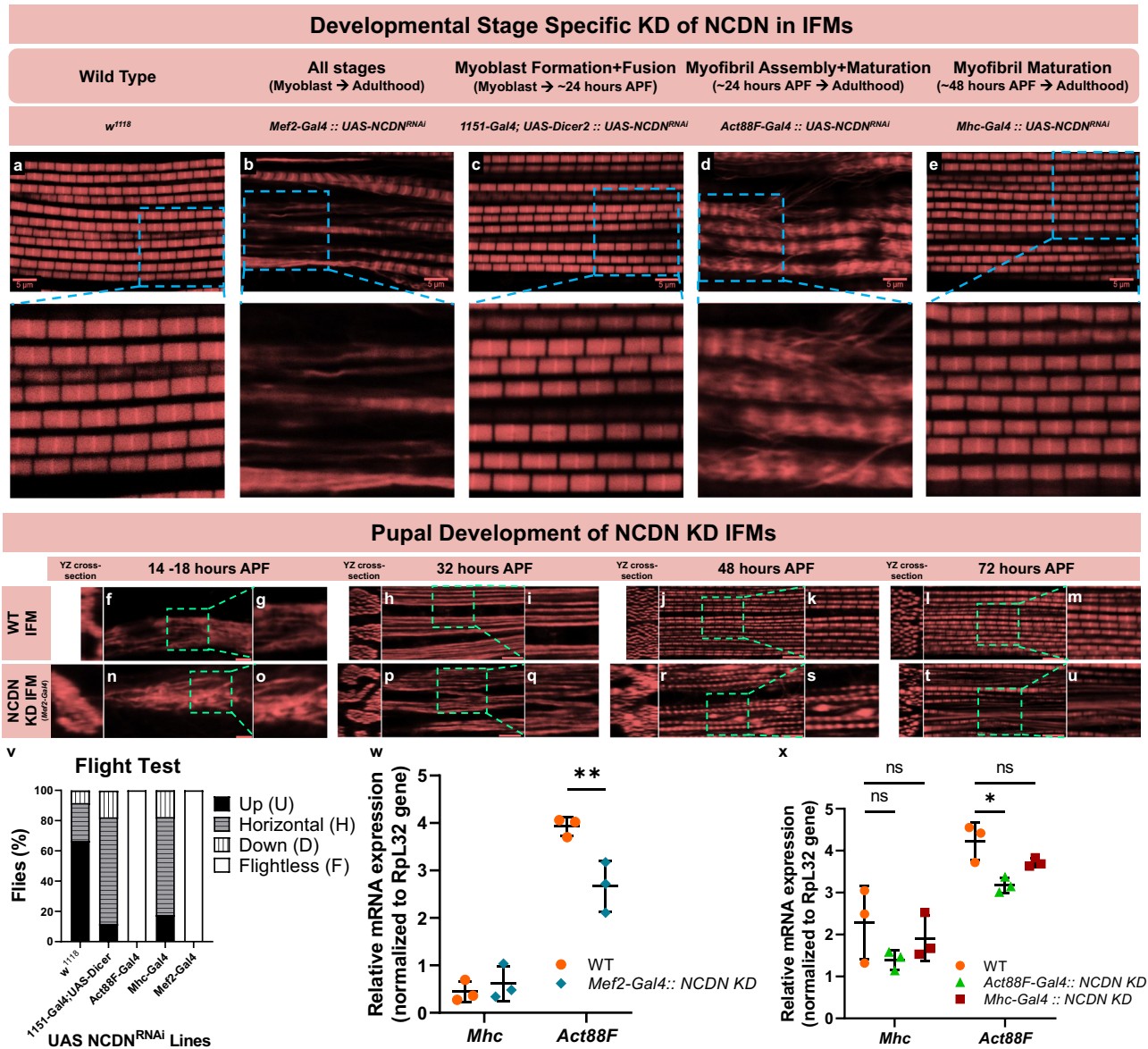

**Fig. 5 Neurochondrin regulates sarcomere branching during myofibril assembly. a–e** Confocal microscopic images of adult indirect flight muscle (IFM) actin after different developmentally timed knockdowns (KD) of neurochondrin (NCDN). Blue dotted lines highlight the region shown in inset (lower panels). **f–u** Confocal microscopy images of IFM actin during pupal development in wild type and *NCDN* KD flies. Cross-section images are shown at the left of each image and dotted lines highlight the zoomed-in regions shown to the right of each image. Representative of cells from three flies. Scale bars: 5 μm. **v** Flight test results in wild type (WT) and different developmentally timed *NCDN* KD flies. **w** IFM actin (Act88F) and myosin (Mhc) relative RNA expression levels in adult WT and *Mef2-Gal4::UAS-NCDN*[RNAi] flies. **x** IFM Act88F and Mhc relative RNA expression levels in adult WT, *Act88F-Gal4::UAS-NCDN*[RNAi], and *Mhc-Gal4::UAS-NCDN*[RNAi] flies. Bars represent overall mean ± SE from three samples (individual points) of 20 pooled thoraces per group. Asterisks (*): Significantly different between groups (**w** two-sided, independent *t* test, **x** one-way ANOVA with Tukey's HSD post hoc test, $P < 0.05$).

that myofibrillar connectivity can be initiated in the IF muscles through either cell-type dependent (e.g., *salm* KD) or cell-type independent mechanisms (e.g., *NCDN* KD and OE), indicating that the *Drosophila* IF muscles offer a unique model for dissecting mechanisms which prevent sarcomere branching.

The conserved nature of myofibrillar network structures across invertebrate and mammalian muscles may also be extended to the cell-type dependent regulators of network connectivity. Muscle-specific loss of transcription factors *exd*, *hth*, *vg*, *salm*, or *H15* each leads to conversion of the IF muscle contractile apparatus from fibrillar to tubular with *exd*, *hth*, and *vg* operating upstream of *salm*[22,35,36] and *H15* downstream of *salm*[25]. Moreover, over-expression of *salm* causes a conversion of the leg muscles to a more fibrillar-like contractile phenotype[22,25], overexpression of

*Exd* and *Hth* together result in the conversion of the jump muscles to a fibrillar phenotype[35], and overexpression of *H15* is capable of returning IF muscles lacking *salm* to a fibrillar phenotype[25]. Based on the lack of myofibrillar networks in the fibrillar IF muscles[20,22,23] (Fig. 1), the presence of myofibrillar networks in each of the three tubular muscles assessed here (Fig. 1), and the initiation of myofibrillar connectivity in the *salm* KD IF muscles (Fig. 3), we propose that each of these regulators of muscle cell type also acts as an inhibitor of myofibrillar network connectivity in IF muscles. Additionally, while the muscle-specific role of *salm* orthologs (Sall1-4) in vertebrates has yet to be reported, orthologs for *exd* (Pbx1-4), *hth* (Meis1-4), *vg* (Vgll1-4), and *H15* (Tbx15) are each known to be involved in the specification of muscle fiber type[53–56]. Thus, both the cell type

dependence of myofibrillar connectivity (Fig. 1) and the cell-type-dependent mechanisms regulating myofibrillar connectivity (Figs. 2, 3) appear to be widely conserved across species.

Transcription factors that regulate muscle cell type often act on the overall design of the cell and influence both the contractile and metabolic machineries[22,35,36,57,58]. Conversely, cell-type independent regulators of the myofibrillar matrix are likely to operate more directly on the contractile apparatus. Indeed, muscle-specific loss of *NCDN* had no effect of the IF muscle expression of cell type regulators Salm and H15 nor on the configuration of mitochondrial networks running parallel in between the myofibrils (Supplementary Fig. 5). Additionally, tracheoles were still present within the *NCDN* KD muscle interior (Supplementary Video 10) as only occurs in fibrillar muscles in a *salm*-dependent manner[59,60], and *NCDN* KD did not cause a switch to the tubular forms of actin and Mhc (Supplementary Fig. 4i, k–s). However, while the majority of the myofibrillar volume in *NCDN* KD IF muscles retained the large, roughly circular profile characteristic of fibrillar muscles (Fig. 4), many thin myofibrillar branches were formed which provided connections between the larger myofibrils. Actin (Fig. 4b, d), myosin filaments (Supplementary Video 10), and z-disk structures (Supplementary Video 12) can each be observed in the thin myofibrillar segments suggesting that the major sarcomeric components are retained by these branches. While there have been few volume electron microscopy studies of *Drosophila* muscle[36] with which to compare our data, the light microscopic images of *NCDN* KD muscles (Supplementary Fig. 4a–f) are reminiscent of previous reports describing IF muscle sarcomere fraying[21,50,61,62]. In fact, loss of *diaphanous (dia)*, an actin polymerization factor from the formin family, has been shown to induce fraying of *Drosophila* IF muscle myofibrils[21]. Interestingly, mammalian orthologs of *dia* (Dia1) and *NCDN* (NCDN) are known to be interacting partners where NCDN can inhibit Dia1-induced neurite outgrowth, albeit not through regulation of actin polymerization activity[40]. While we find here that loss of *NCDN* does not result in a change in Dia RNA expression (Supplementary Fig. 4j), we cannot rule out whether *NCDN* influences the actin polymerization activity of dia. However, we do find that loss of *NCDN* results in a reduction in Act88F, Act79B, and Actn RNA expression without a change in Mhc expression, thereby altering the actin-myosin balance previously shown to regulate the formation of the fibrillar IF muscle myofibrils[49]. Moreover, this loss of actin-related transcripts in *NCDN* KD muscles is consistent with previous work showing an up-regulation of NCDN and several actin cytoskeletal transcripts in an Mhc mutant hypercontraction fly model[41]. Determining whether *NCDN* interacts directly with actin, myosin, *dia*, or other proteins involved in sarcomere fraying[50,61] to mediate myofibrillar connectivity in *Drosophila* muscle will be critical to the identification of the molecular mechanisms acting directly on the sarcomere to induce branching in a cell-type-independent manner.

The forces generated by contraction of the muscle sarcomere are critical for the coordinated movement of animals from insects to humans. Force transmission down the longitudinal axis of the muscle fiber through sarcomeres arranged in series has long been considered the primary means by which sarcomeric forces arrive at the muscle tendon and ultimately act upon the skeleton to create movement[8–10,16,63,64]. Transmission of force laterally from the sarcomere to the sarcolemma and extracellular matrix which is then propagated to the tendon is now also appreciated, at least in vertebrate muscles, although the specific structures that permit lateral force transmission have remained elusive[65–68]. The presence of a singular, mesh-like myofibrillar matrix linked together through branching sarcomeres shown previously in mammals[11] and amphibians[16] and now here in invertebrates provides a direct

structural pathway for lateral force transmission across the width of the muscle cell. However, additional structural components such as the cytoskeleton[65–68] and the costamere[69–71] are still needed for the transmission of forces from sarcomeres located at the periphery of the cell to the sarcolemma and extracellular matrix. Thus, elucidating the structural and functional interactions among the active (myofibrillar matrix) and passive (cytoskeleton/costamere) force transmitting elements of the muscle cell will be of particular importance, especially under pathological conditions such as in muscles lacking the costameric protein dystrophin[72,73]. The large dynamic range of myofibrillar network connectivity shown across muscle types here combined with the widely available genetic tools suggests that *Drosophila* offers a promising model system for investigating the integrated nature of muscular force transmission[74,75]. However, several technical challenges remain to be overcome in order to directly assess the impact of sarcomere branching and myofibrillar network formation on force transmission. Though we have identified three different genes here (*salm*, *H15*, *NCDN*) which regulate the frequency of sarcomere branching by two different mechanisms (cell-type dependent and independent), misexpression of each of these genes alters much more than just the configuration of the myofibrillar network. In addition to *salm* KD and *H15* KD previously being shown to alter mitochondrial network morphology in the IF and TDT muscles, respectively[25,36], we find that despite not altering the parallel arrangement of IF muscle mitochondria (Supplementary Fig. 5), many *NCDN* KD IF muscle mitochondria have electron-dense inclusions (Supplementary Video 10) which are indicative of reduced mitochondrial capacity for contractile support[76]. Non-contractile defects can generally be minimized by performing contractile experiments on permeabilized muscles[77] or using isolated linear myofibril segments[78]. However, to assess contractile properties in isolated myofibril branches will require the development of an apparatus that can connect to three separate myofibril ends (e.g., Supplementary Fig. 2b), while direct assessment of the functional impact of sarcomere branching using either permeabilized muscles or a yet to be developed isolated branching myofibril apparatus will require models where sarcomere branching, but no other contractile properties have been altered. Here, the *salm*, *H15*, and *NCDN* KD muscles have either frequent streaming Z-disks (*NCDN*) or holes in the majority of Z-disks (*salm* and *H15*) suggesting a major contractile defect consistent with the lack of flight[25] (Fig. 5v) and jump[25] ability in these flies. Thus, until a specific regulator of sarcomere branching is identified, there remains a great need for the development of novel biophysical- or imaging-based approaches to assess non-linear contractile forces approaching sarcomeric resolution.

## Methods

**Drosophila strains and genetics**. Genetic crosses were performed on yeast corn medium at 22 °C. W[1118] were used as controls for the respective genetic backgrounds. *Mef2-Gal4* (III chromosome, BS# 27390), *1151-Gal4*[79] (I, gift from Upendra Nongthomba);*UAS-Dicer2* (BS# 24650), *Act88F-Gal4* (III, BS# 38461), and *Mhc-Gal4*[80] (III, gift from Upendra Nongthomba) were used to drive muscle-specific gene knockdown of respective genes. *UAS-mito-GFP* (II, BS# 8442) was used as a UAS control. RNAi lines for knockdown of *salm* (*UAS-salm RNAi*, V101052), *H15* (*UAS-H15 RNAi*, V28415), and *Neurochondrin (NCDN)* (*UAS-NCDN RNAi*, V109002) were purchased from the Vienna Drosophila Resource Center. Mhc-GFP-weeP26 transgenic flies[81] were a kind gift from Maxim Frolov. *UAS-NCDN* (Neurochondrin) transgenic flies were generated using the plasmid containing BDGP Tagged *NCDN* ORF (UFO10226; obtained from DGRC). We confirmed the fragment size by performing digestion with EcoRI and further amplified the fragment by PCR and verified by sequencing. The *UAS-NCDN* transgenic flies were generated using phiC31-mediated integration into the *attP2* landing site, and the injection of embryos was performed by Bestgene Inc (USA). All other stocks were obtained from the Bloomington (BS#) Drosophila Stock Center. All chromosomes and gene symbols are as mentioned in Flybase (http://flybase.org).

**Muscle preparation**. Muscles from 2–3-day-old flies were dissected on the standard fixative solution (2.5% glutaraldehyde, 1% formaldehyde, and 0.1 M sodium cacodylate buffer, pH 7.2) and processed for FIB-SEM imaging as described previously[26]. Muscles were then transferred to the fresh fixative Eppendorf tube and fixed overnight at 4 °C. Samples were washed in 0.1 M cacodylate buffer for 10 min, three times. Later, fixed tissues were immersed in a solution of 1.5% $K_3[Fe(CN)_6]$ in 0.1 M cacodylate buffer with 2% $OsO_4$ on ice for 1 hr. Samples were washed three times with distilled water for 10 min each and were then incubated in filtered thiocarbohydrazide (TCH) solution for 20 min at room temperature. This was followed by three washes with water for 10 min each, then incubation for 30 min in 2% $OsO_4$ on ice. Incubated samples were washed with distilled water for 10 min, three times each. Tissues were transferred to 1% uranyl acetate at 4 °C overnight. Furthermore, samples were washed with distilled water three times for 10 min, then incubated with Walton's lead aspartate solution at 60 °C for 30 min. Samples were again washed with warm distilled water and then dehydrated with increasing order of alcohol percentage (20%, 50%, 70%, 90%, 95%, 100%, 100%) at room temperature. Later, samples were transferred to the freshly made 50% epoxy resin in alcohol and incubated for 4–5 h at room temperature and then incubated in 75% epoxy overnight at room temperature. Samples were transferred to freshly prepared 100% Epon812 resin and incubated for 1 h, then transferred to fresh 100% Epon812 resin for another 1 h incubation, and finally transferred to 100 % Epon812 for a 4 h incubation. Tissues were then transferred to the stub with as little resin as possible and incubated for polymerization on the stub for 48 h at 60 °C.

**FIB-SEM imaging**. FIB-SEM images were acquired using a ZEISS Crossbeam 540 with ZEISS Atlas 5 software (Carl Zeiss Microscopy GmbH, Jena, Germany) and collected using an in-column energy selective backscatter detector with a filtering grid to reject unwanted secondary electrons and backscatter electrons. Images were acquired at 1.5 kV, 2 nA probe current at the working distance of 5.0 mm. FIB milling was performed at 30 keV, 700 pA current, and 10 nm thickness. Image stacks within a volume were aligned using Atlas 5 software (Fibics Incorporated, Ontario, Canada) and exported as TIFF files for analysis. Voxel size was set at $10 \times 10 \times 10$ nm.

**Image segmentation**. Raw FIB-SEM image volumes were rotated in 3D so that the XY orientation represented the cross-section of the muscle. The image stack volume was imported into the TrakEM2 plugin in ImageJ. A myofibril was chosen in the first slice and cross-sections of sarcomeres were traced along the longitudinal axis of the muscle. Tracings were then stitched together using the interpolation feature in TrakEM2. When sarcomeres branched or merged during tracing via the presence or absence of the sarcoplasmic reticulum/t-tubules, cross-sections were traced immediately before and after the connection point to ensure accurate interpolation. For example, when arriving at a sarcomere that splits into two, the previously traced segment prior to the branch point was continued along one part of the branch using the same label and a new label was created to trace the other branched segment. In the case of a merging event, the method applied was in reverse from that of a splitting event. When two sarcomeres merged at a connection point, only one of the two labels used to keep track of the two previously traced segments were chosen to trace downstream segments. Therefore, each myofibrillar segment of the matrix was identified with a different label. The accuracy of interpolated tracings was assessed by overlaying the exported tracings from TrakEM2 with the raw FIB-SEM file. A multi-color representation of the myofibrillar matrix was obtained following a series of steps. First, a myofibrillar mask with myofibrillar pixels assigned a value of 1.0 and background pixels assigned as null (NaN or ignored in all mathematical analyses) values were created by dividing the myofibrillar image by itself using the Image Calculator tool while choosing 32-bit (float) result. The myofibrillar mask image was then multiplied by the raw 32-bit FIB-SEM data. Finally, the resulting grayscale image stack was then merged with the myofibrillar labels file, with each segment shown as a different color.

For analysis of the overall contractile, mitochondrial, and SRT volumes, FIB-SEM volumes were segmented in semi-automated fashion using Ilastik[82] machine learning software as described previously[26]. Briefly, FIB-SEM image volumes were binned to 20 nm in ImageJ, saved as 8-bit HDF5 files, and imported into Ilastik. Pixel classification training using all available features was performed for the contractile apparatus, mitochondria, SRT, and all other pixels and exported as 8-bit probability files. The resultant HDF5 files were imported back into ImageJ and binarized using a 50% threshold. The binary structures were then filtered using the Remove Outliers plugin in ImageJ using a 3–10 pixel radius and 1.5–2 standard deviations and the number of resultant pixels was counted for each structure and compared to the total number of cellular pixels.

For analyses relating to the sarcolemmal boundary, the sarcolemma was manually traced using the TrakEM2 plugin for ImageJ.

**Quantitative analysis of myofibrillar networks**. Raw FIB-SEM image volumes were rotated in 3D so that the muscle cell's cross-section was visible. Every sarcomere in parallel in the first image within the dataset was numbered and subsequently tracked along the longitudinal axis for sarcomere connectivity. Only

sarcomeres that stayed within the field of view of our datasets were used for analysis. We define branching as interconnections between nearby sarcomeric structures. Analyses were done per sarcomere and as such, the number of times that a sarcomere split, merge, or had no event(s) were counted. Tracking proceeded down the length of the muscle with the portion of the myofibrillar structure which best maintained the same relative location within the muscle fiber (i.e., distance from the cell membrane) as the original numbered sarcomere from the first image.

Myofibril cross-sectional area (CSA) and circularity were measured from the traced structures by converting them to binary images and running the Analyze Particles plugin in ImageJ for each slice throughout the volume. The volume-weighted average myofibril CSA was calculated for the wild type and *NCDN* KD IF muscles by multiplying the CSA of each myofibril by its percentage of the total CSA for all myofibrils and then summing the resultant value for all myofibrils.

Analyses of the sarcomere branching position relative to the sarcolemma were performed by importing both the traced sarcolemma boundary and the sarcomere branch points into Imaris 9.7 (Bitplane) and running a surface to surface distance transform analysis.

**Image rendering**. 3D animations of the myofibrillar networks were created in Imaris using the horizontal and vertical rotation features and/or the clipping tool. Additional 3D renderings were created using the Volume Viewer plugin in ImageJ.

**Real-time quantitative PCR**. Total RNA was extracted from the thorax muscles ($n = 20$ per trial) from wild-type and *NCDN* KD flies using the TRIzol™ Plus RNA Purification Kit (A33254) according to the manufacturer's instructions. DNA digestion was carried out using TURBO DNA-free™ Kit (AM1907). First-strand synthesis was performed on 500 ng of total RNA using SuperScript™ VILO™ cDNA Synthesis Kit (Thermo Fisher Scientific, 11754050) according to the manufacturer's instructions. Concentrated cDNA from all samples was used to generate standard curves. Amplification was detected using the QuantStudio Real-Time PCR System (Thermo Fisher Scientific). The measures were normalized to *Rpl32* reference gene. All reactions were performed in triplicates. Probes used: *Neurochondrin* (Assay ID Dm02137509_s1), *RpL32* (Assay ID Dm02151827_g1), *Act88F* (Assay ID Dm02362815_s1), dia (Assay ID Dm01811204_m1), Actn (Assay ID Dm01821076_m1), Act79B (Assay ID Dm01823319_gH), sls (Assay ID Dm01825768_g1), bt (Assay ID Dm01808089_m1), tsr (Assay ID Dm01842418_g1), fln (Assay ID Dm01823176_g1), TpnC4 (Assay ID Dm01815264_m1), and Mf (Assay ID Dm02140561_g1) (Thermo Fisher Scientific).

**Immunofluorescent analysis**. IF muscles were dissected from the thorax in 4% formaldehyde and were processed as described previously[83]. Briefly, thoraces with IF muscles were fixed in 4% formaldehyde for 2 h at room temperature on a rotor. Samples were washed three times with PBS + 0.03% Triton X-100 (PBSTx) for 15 min and blocked for 2 h at room temperature or overnight at 4 °C using 2% bovine serum albumin (Sigma). Samples were incubated with and without respective primary antibody (Ab) at 4 °C overnight and later washed three times for 10 min with PBSTx and incubated for 2.5 h in respective secondary Ab at 25 °C or overnight at 4 °C. Samples were incubated for 40 min with Phalloidin TRITC (2.5 μg/ml) (P1951, Sigma, USA) and mounted using Prolong Glass Antifade Mountant with NucBlue stain and incubated for 20 min. Images were acquired with a ZEISS 780 confocal microscope and processed using the ImageJ and ZEN software (version 3.2.0.115) respectively. Antibodies used for the staining: Rabbit anti-salm-1 (1:500, gift from Dr. Tiffany Cook[84]), Rabbit anti-nmr1 (H15) (1:200, gift from Dr. James B. Skeath[85]), and Alexa-Fluor-488-labeled anti-rabbit IgG (1:500, Cat# A32731, Thermo Fisher Scientific).

**Statistical analysis**. Statistical analyses were conducted using Excel 2016 (Microsoft, Redmond, WA) and Prism 9.0.0 (GraphPad, San Diego, CA). When appropriate, data points were overlayed with their corresponding summary plots to create a SuperPlot. All comparisons of means between wild-type muscles of the indirect flight, jump, direct flight, and leg muscles were performed by one-way analysis of variance. Brown–Forsythe tests were performed to determine the equality of group variances. When the variances were determined to be different, a Dunn's multiple comparisons test was performed. When variances were determined to not be different, a Tukey's HSD post hoc test was performed. Differences between the gene knockdowns and corresponding wild-type muscles were evaluated on the means from each dataset using a two-sided independent $t$ test for data that passed a Shapiro–Wilks normality test. A Wilcoxon signed-rank test was performed in lieu of the Shapiro–Wilks normality test if insignificant. Statistical significance was set at a $P$ value of 0.05.

**Reporting summary**. Further information on research design is available in the Nature Research Reporting Summary linked to this article.

## Data availability

Several raw FIB-SEM datasets are freely available at https://doi.org/10.5281/zenodo.5796264. Additional raw data used in this work are available upon reasonable request. Source data are provided with this paper.

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

## Acknowledgements

Present study was supported by the Division of Intramural Research of the National Heart, Lung, and Blood Institute (NHLBI, 1ZIAHL006221 to B.G.) and the National Institute of Arthritis and Musculoskeletal and Skin Diseases (NIAMS). The authors gratefully acknowledge Dr. Eric Lindberg for his assistance with FIB-SEM preparation and acquisition in the initial stage of the project and Shree Chaitranjali Yadla for helpful discussions regarding developmental *neurochondrin* expression.

## Author contributions

P.T.A., P.K., C.K.E.B., and B.G. designed; P.K and Y.S. prepared samples; Y.S. and C.K.E.B. imaged samples; and P.T.A., T.B.W., and B.G. analyzed images for all electron microscopic studies. Y.Z. and P.K. generated *neurochondrin* overexpression fly line. P.K. and B.G. designed and P.K. performed and analyzed all light microscopic experiments. P.T.A. and B.G. wrote the manuscript draft and P.T.A., P.K., Y.Z., T.B.W., Y.S., C.K.E.B., and B.G. edited and approved the manuscript.

## Funding

## Competing interests

The authors declare no competing interests.
