## [Peer Review File · Nature Communications]

Regulation of the Evolutionarily Conserved Muscle Myofibrillar Matrix by Cell Type Dependent and Independent MechanismsEditorial Note: Parts of this Peer Review File have been redacted as indicated to remove third-party material where no permission to publish could be obtained.

REVIEWER COMMENTS

Reviewer #1 (Remarks to the Author):

Ajayi et al perform an interesting study that demonstrates the muscle specific branching of myofibrils in drosophila. Further to this they investigate the KD and expression of various genes to demonstrate the importance of neurochondrin, H15 and salm in myofibrillar branching regulation. This work is interesting, novel and contains a lot of work in the analysis of the 3D EM data which is demonstrated in fantastic images and videos. I have a few suggestions and question to the authors that I hope will help improve the manuscript further but I certainly think this work should be published in Nature communications following revision.

- The introduction ends with what appears to be a summary of the main findings. This is an unusual approach and I would suggest instead using the final paragraph to summarise the aim of the work rather than the findings.
- Figure 1f – The figure legend needs to be clearer. Is this percentage of sarcomeres within each myofibril that are branched?
- The authors have shown variation in the degree of branching dependent on the muscle type and state that there is more branching in the more oxidative muscles than the more glycolytic muscles. Can the authors determine if this is the muscle type or the fibre type that determines this?
- How did the authors determine that branching is more common in the periphery of the fibre? Was it quantified or is it an observation, could you please clarify this?
- The authors look at the degree of sarcomere branching in relation to myofibrillar and mitochondrial content as an indicator of whether branching is related to force production or energy conversion. However, they haven't looked at force production directly, can they provide evidence that myofibrillar content is related to force production? Or even better can they measure force production? This is important to understand the functional relevance of sarcomere branching.
- The authors state that the H15 KD data suggests that H15 regulates myofibrillar branching in the TDT muscles by converting them towards a leg muscle phenotype. This should be rephrased since the conversion upon knockdown suggests the opposite and that H15 is important for the TDT phenotype and that a lack of H15 leads to the leg muscle-like phenotype. Did the authors look to see whether there is a difference in expression of H15 in wildtype leg and TDT muscles?
- When the authors filtered the proteomic data and identified NCDN based on the expression pattern in the different muscles were there other candidates that were identified and if so how were these ruled out?
- Figure 4K shows a gradation of myofibril shape through the different WT muscles, it would be interesting and a valuable addition to the paper to see this alongside the level of NCDN expression or protein levels as well as the other genes (H15, Salm etc) identified as playing a role to better visualise their respective contributions. Particularly since the authors propose that the other genes play an inhibitory role in myofibrillar branching in the discussion.
- The authors discuss the role of myofibrillar branching and a single myofibrillar matrix in the generation of force. Can they compare force generation in the same muscles from their NCDN KD v WT drosophila to demonstrate the relationship of the branching with force generation and substantiate this?
- In the results the authors mention quantifying levels of mRNA but not that this was by qPCR which is only clear when the methods are read, it would be helpful to include this on first mention in the results.
- The methods section does not provide information on whether no primary controls were included in staining experiments for salm and nmr, were these completed. Further did the authors check if the salm antibody appeared specific by assessment in salm KD vs wild type?

Reviewer #2 (Remarks to the Author):

NCOMMS-21-39230

The manuscript by Ajayi et al describes a fascinating analysis of the newly-discovered myofibrillar matrix - the anastomoses and cross-linking of parallel myofibrils within individual muscle fibers - in the *Drosophila* system. The authors set out to determine the existence and distribution of this phenomenon in adult insect muscles, and evaluate the potential contributions of specific genes to regulating the existence of the matrix.

The authors find that, while the fibrillar indirect flight muscles do not show existence of a myofibrillar matrix, the TDT muscles show it to a certain extent, and other adult muscles to a greater extent, particularly the leg muscles. These findings rely exclusively on focused ion beam SEM, and a modest concern is that it might be valuable to also observe these events using more traditional EM approaches, such as TEM of longitudinal sections. Such data would add experimental robustness to the very impressive data that are already presented.

In the second part of the manuscript, the authors determine if knockdown of candidate genes that are known or thought to modulate muscle fiber identity also correspondingly transform the myofibrillar matrix. Indeed knockdown of *salmon* in the flight muscles, which was previously shown to result in a transformation of IF muscles to tubular muscles, also results in the appearance of a matrix in the transformed IF muscles. Similarly, knockdown of H15, not previously shown to regulate muscle fiber fate, also causes acquisition of leg a muscle-like network in the TDT. Finally, knockdown of *Neurochondrin* causes the appearance of the myofibrillar matrix in the IF muscles.

These studies are significant in showing the existence of the myofibrillar matrix in insect muscles, its diverse occurrence across different muscle types, and some aspects of its genetic control. However the genetic approach was not as thorough: while the knockdown data were compelling, there were no corresponding gain of function approaches, such as determining if over-expression of *neurochondrin*, H15 or *salmon* would have effects opposite to their loss of function phenotypes.

The fraying and merging of adjacent myofibrils as noted by the authors in the NCDN knockdowns are indeed interesting and not unique. In addition to the recent papers the authors cite, it is notable that haploinsufficiency for several myofibrillar protein genes can also result in fraying and merging of adjacent myofibrils (see for example Beall et al 1989, *Genes Dev*). Since the authors do not investigate the cellular mechanisms that control branching, is it possible that the NCDN knockdown simply dysregulates actin or myosin assembly?

Overall this is an interesting study that enables the analysis of a novel but significant phenomenon in a genetically tractable organism. However, there are weaknesses with the work in its current state due to a lack of genetic rigor and a failure to provide insight into the underlying molecular control of myofibrillar matrix formation.

Additional comments/questions.

1. Did the authors focus upon a specific direct flight muscle for their analyses, and if so which muscle was this?
2. Line 196: should "towards" be "away from"?
3. Figures 2, 3, and 4 lack comparable control images, since these images are at a different

magnification from those shown in Figure 1.

4. Figures 2-4, please could that authors draw boxes around each graph as it is difficult to see where one graph ends and the next one starts.

5. Line 261-262, I do not see any evidence that would support the description of H15 as a "fibrillar muscle specification" factor.

6. Figure 4, panels c and d. These should be at the same magnification.

Reviewer #3 (Remarks to the Author):

Glancy and colleagues describe the branching pattern of myofibrils in different *Drosophila* muscles by FIB-SEM. They have previously done similar work in vertebrate muscles and now show that branching is conserved in insect muscles (TDT and leg).

Overall, the imaging and the data analysis is very solidly done. Nevertheless, one does not learn much from this paper.

H15 and the adjacent gene midline encode closely related and recently duplicated T-box transcription factors that are co-regulated and have similar functions in *Drosophila*. Why is this not mentioned anywhere? Does a midline KD have a similar phenotype?

The salm phenotype is not surprising, given that salm is upstream of H15 and is the master regulator of IFM fate.

The neurochondrin phenotype is exciting, but somewhat haphazardly introduced and not necessarily convincing. I find it curious how the authors measure cross-sectional area. To me the most apparent phenotype of neurochondrin KD appears to be myofibrils with a bigger diameter (bigger cross-sectional area), which is quite apparent from 4b and d. Only by counting every "wisp" of actomyosin separating from a myofibril as a myofibril do they get the smaller diameter. The phenotype they describe is correct, but perhaps the interpretation could also be that the main phenotype of neurochondrin KD is an increase of myofibril cross-sectional area, which eventually leads to instability and fraying of myofibrils. In summary, this phenotype looks different from the branching documented in Figure 1 (branches are much thinner and more frequent), making me doubt that it is the same phenotype.

A more descriptive paper without the neurochondrin phenotype might be more valuable for the community, in simply documenting the fact of myofibril branching in some muscle types in *Drosophila*.

Reviewer #1 (Remarks to the Author):

Ajayi et al perform an interesting study that demonstrates the muscle specific branching of myofibrils in drosophila. Further to this they investigate the KD and expression of various genes to demonstrate the importance of neurochondrin, H15 and salm in myofibrillar branching regulation. This work is interesting, novel and contains a lot of work in the analysis of the 3D EM data which is demonstrated in fantastic images and videos. I have a few suggestions and question to the authors that I hope will help improve the manuscript further but I certainly think this work should be published in Nature communications following revision.

We thank the reviewer for their thoughtful suggestions and questions which have helped strengthen the manuscript.

- The introduction ends with what appears to be a summary of the main findings. This is an unusual approach and I would suggest instead using the final paragraph to summarise the aim of the work rather than the findings.

We agree that including a summary at the end of the introduction is not the classical way of constructing a manuscript. However, it is becoming more commonplace these days, especially in certain journals. In fact, we surveyed 20 Nature Communications articles which were published on November 8th, 2021 and found that 17 of 20 contained summary information at the end of the introduction. Thus, in order to stay in line with formatting practices for this journal, we have left the summary information but have now also added two opening sentences describing the goal and hypothesis driving this study.

- Figure 1f – The figure legend needs to be clearer. Is this percentage of sarcomeres within each myofibril that are branched?

We thank the reviewer for this comment and have now clarified that it is the percentage of sarcomeres per myofibril that are branched.

- The authors have shown variation in the degree of branching dependent on the muscle type and state that there is more branching in the more oxidative muscles than the more glycolytic muscles. Can the authors determine if this is the muscle type or the fibre type that determines this?

We thank the reviewer for this comment. Here, the jump, direct flight, and leg muscles are all tubular fiber types with the same primary actin isoform (Act79B) while the fibrillar flight muscle has a different primary isoform (Act88F). Thus, it appears that it is the overall muscle cell type, rather than the contractile fiber type which determines branching frequency. We have now made mention of this in the results section when discussing Figure 1.

- How did the authors determine that branching is more common in the periphery of the fibre? Was it quantified or is it an observation, could you please clarify this?

We thank the reviewer for catching the omission of this information from the methods section. This data was quantified in Supplemental Figure 3 included at the bottom of the original manuscript. A distance transform analysis was performed to measure the distance between each sarcomere branch point within a cell and the sarcolemma, and this information has now been added to the methods at the end of the Image Segmentation and Image Analyses sections.

- The authors look at the degree of sarcomere branching in relation to myofibrillar and mitochondrial content as an indicator of whether branching is related to force production or energy conversion. However, they haven't looked at force production directly, can they provide evidence that myofibrillar content is related to force production? Or even better can they measure force production? This is important to understand the functional relevance of sarcomere branching.

We thank the reviewer for this comment. Generally speaking, the amount of machinery available to perform a given task is positively correlated with the overall capacity to perform that task. This is why force production measurements are typically normalized to the amount of muscle, mitochondrial function measurements are normalized to mitochondrial content, and enzyme assays are normalized for enzyme content, for example. For force production measures, usually muscle mass or cross-sectional area (CSA) are used as the normalization factor, and it has been shown that myofibrillar CSA is a better normalization factor than muscle CSA because it more directly accounts for the actual machinery that leads to force production (Taylor and Kandarian, J Appl Physiol, 1994). We have now included this reference when introducing the myofibrillar content measurement.

There are relatively few papers which compare force production across different *Drosophila* muscle types and they have generally been restricted to indirect flight and jump muscles. These data show that jump muscles generate much greater force than indirect flight muscles (Swank, Methods, 2012 and Peckham et al. J of Muscle Research Cell and Motility, 1990) consistent with the much greater myofibrillar content. However, based on the myofibrillar contents and sarcomere branching frequencies across all four wild type muscles in Figure 1, we concluded that branching frequency was not related to myofibrillar content and thus, force production capacity. This lack of relationship is also supported in the mammalian system where force production characteristics have been widely described with muscle power and total force production being highest in fast-twitch, followed by slow-twitch, and then cardiac muscles. However, branching frequency is highest in slow-twitch followed by cardiac and then fast-twitch muscles (Willingham et al. Nat Comms, 2020). Thus, there does not appear to be a direct relationship between maximal force production and sarcomere branching frequency.

- The authors state that the H15 KD data suggests that H15 regulates myofibrillar branching in the TDT muscles by converting them towards a leg muscle phenotype. This should be rephrased since the conversion upon knockdown suggests the opposite and that H15 is important for the TDT phenotype and that a lack of H15 leads to the leg muscle-like phenotype. Did the authors look to see whether there is a difference in expression of H15 in wildtype leg and TDT muscles?

We thank the reviewer for catching this error in the text which we have now corrected. "Overall, these data suggest that H15 regulates the connectivity of myofibrillar networks in the tubular TDT muscle by preventing these muscles from taking a leg muscle phenotype."

Also, H15 is expressed more than five-fold greater in TDT than leg muscles based on our mass spectrometry analyses that identified H15 as a muscle type specification factor in Drosophila (Katti et al. bioRxiv, 2021 doi: <https://doi.org/10.1101/2021.09.30.462204>).

- When the authors filtered the proteomic data and identified NCDN based on the expression pattern in the different muscles were there other candidates that were identified and if so how were these ruled out?

We thank the reviewer for this question. The original purpose of the proteomic screen from Katti et al. (bioRxiv, 2021) was to identify factors related to fibrillar or tubular contractile types, parallel or grid-like mitochondrial networks, or Salm expression. 142 candidate proteins were identified and 10 were misexpressed and at least preliminarily screened for mitochondrial and contractile phenotypes by light microscopy (knockdown and overexpression of 5 transcription factors, including H15 and salm, are included in Katti et al.). We found that sarcomere branching frequency (leg>TDT>IFM and salm KD>IFM) from Figure 1 here is inversely correlated with Salm expression (IFM>TDT>leg and IFM>salm KD) based on Schonbaur et al (Nature, 2011), Bryanstev et al (Dev Cell, 2012) and Katti et al (bioRxiv, 2021), so we focused on the list of proteins negatively associated with Salm and relaxed the relative expression threshold from 50% to 25% which expanded our list from 50 to 193 proteins. We then sorted the list for the proteins which were most upregulated in Salm KD IF muscles. NCDN is one of 17 proteins that were more than 10-fold upregulated by Salm KD. Several, but not all, of these 17 were misexpressed and screened by light microscopy, and the NCDN KD phenotype appeared to cause sarcomere branching, so we proceeded to the more detailed FIB-SEM analyses shown here.

- Figure 4K shows a gradation of myofibril shape through the different WT muscles, it would be interesting and a valuable addition to the paper to see this alongside the level of NCDN expression or protein levels as well as the other genes (H15, Salm etc) identified as playing a role to better visualise their respective contributions. Particularly since the authors propose that the other genes play an inhibitory role in myofibrillar branching in the discussion.

We thank the reviewer for this suggestion. We have now added Supplementary Table 1 which summarizes the expression level of Salm, H15, and NCDN as well as sarcomere branching frequency across muscle types.

- The authors discuss the role of myofibrillar branching and a single myofibrillar matrix in the generation of force. Can they compare force generation in the same muscles from their NCDN KD v WT drosophila to demonstrate the relationship of the branching with force generation and substantiate this?

We thank the reviewer for this question. Our goal here was to determine whether myofibrillar connectivity is evolutionarily conserved in Drosophila muscles which would then open the door to the wide array of transgenic fly tools to be used for dissecting the mechanisms of sarcomere branching. This goal has now been added to the introduction. As such, we were able to identify three genes which regulate myofibrillar connectivity by two different mechanisms (cell-type dependent/independent). However, we would not expect comparisons of force production between the transgenic and respective wildtype muscles here to be reflective of the direct impact of

myofibrillar connectivity for several reasons. First, while the salm, H15, and NCDN KD flies are all viable with intact muscles, they are all flightless indicating defective muscle function. Second, Katti et al. (bioRxiv, 2021) showed reduced jumping ability in the salm KD (2+ fold) and H15 KD (5+ fold) flies compared to wild types indicating reduced TDT muscle function. Third, z-disk structure (see Supplementary Movies 5,9,12-13) is largely disrupted in the salm KD IFMs (holes), H15 KD TDT (holes), and NCDN KD IFMs (streaming) consistent with defective contractile function. Fourth, there are many autophagosomes present in between the myofibrils of the NCDN KD IFMs (see Supplementary Movie 10) indicative of muscle pathology. Finally, while the mitochondrial networks in NCDN KD IFMs remain located in parallel to the myofibrils, the internal structure of the mitochondria features many electron dense inclusions (see Supplementary Movie 10) which is indicative of dysfunctional mitochondria. Thus any potential differences in contractile function of single cells or whole muscles would be greatly confounded by the many cellular changes in these transgenic flies. Additionally, while isolated myofibril preparations can be used to assess contractile performance without influence from other cellular factors, branching myofibrils have more than two ends, and are not compatible with an isolated myofibril measurement apparatus. Moreover, isolated myofibril assays would not account for lateral force transmission which is the component sarcomere branching would be expected to contribute to most. The best approaches to assess the functional significance of sarcomere branching are likely to either develop a computational model capable of simulating force transmission through contractile networks of different geometries, to develop new fluorescent probes which are capable of assessing force within different regions of the contractile apparatus during muscle contraction, or to develop a new myofibril force production apparatus which can assess branched structures. Unfortunately, each of these approaches require significant undertaking which is beyond the scope of this paper. We have now added some discussion relating to the challenges to be overcome in order to directly assess the impact of sarcomere branching on force transmission.

- In the results the authors mention quantifying levels of mRNA but not that this was by qPCR which is only clear when the methods are read, it would be helpful to include this on first mention in the results.

We thank the author for this suggestion and have now mentioned in the results that mRNA for NDCN and the newly added contractile genes was measured by qPCR.

- The methods section does not provide information on whether no primary controls were included in staining experiments for salm and nmr, were these completed. Further did the authors check if the salm antibody appeared specific by assessment in salm KD vs wild type?

We thank the reviewer for this question. We have now indicated in the methods that samples were incubated both with and without primary antibodies as a control. The sensitivity of the salm and H15 antibodies to KD and OE are shown below in three supplemental figures from Katti et al. (bioRxiv, 2021).

Supplementary Fig. S7. Salm expression in *Drosophila* muscles. (a-d) Wildtype flight muscles (IFM), (e-h) jump muscles, and (i-l) leg muscles stained for F-actin (phTRITC), nuclei (DAPI), and Salm antibody. (m-p) *salm* KD IFM showing decreased expression of Salm in nuclei (DAPI). (q-t) Leg muscles with *salm* OE stained for F-actin (phTRITC), nuclei (DAPI), and Salm antibody showing increased Salm expression in the nuclei (Scale Bars: 5 μ m for all).

Supplementary Fig. S15. *H15* is downstream of *salm* in the fiber type specification pathway. (a-d) Wildtype IFM stained for mitochondria (mito-gfp), H15 antibody, and nuclei (DAPI) showing H15 expression in the nuclei. (e-h) *H15* KD IFM showing decreased expression of H15. (i-l) Wildtype IFM stained for mitochondria (mito-gfp), Salm antibody, and nuclei (DAPI) showing

Supplementary Fig. S18. H15 overexpression in *salm* knock down background. (a-d) Wildtype fibrillar flight muscles (IFMs) stained for F-actin (phTRITC), nuclei (DAPI), and H15 antibody showing H15 expression in the nuclei. (e-h) *H15* OE IFM showing increased expression of H15. (i-l) *H15* OE; *salm* KD IFM showing overexpression of H15 (Scale Bars: 5 μ m). (m,n) Quantification of fluorescence intensity of (m) H15 antibody staining. Each point represents value for each dataset. Bars represent mean \pm SD. Significance determined as $p < 0.05$ from one way ANOVA with Tukey's (*, $p \leq 0.05$; **, $p \leq 0.01$; ***, $p \leq 0.001$; ****, $p \leq 0.0001$; ns, non-significant).

Reviewer #2 (Remarks to the Author):

NCOMMS-21-39230

The manuscript by Ajayi et al describes a fascinating analysis of the newly-discovered myofibrillar matrix - the anastomoses and cross-linking of parallel myofibrils within individual muscle fibers – in the *Drosophila* system. The authors set out to determine the existence and distribution of this phenomenon in adult insect muscles, and evaluate the potential contributions of specific genes to regulating the existence of the matrix.

We thank the reviewer for these comments and for their critical insights which have greatly improved the manuscript.

The authors find that, while the fibrillar indirect flight muscles do not show existence of a myofibrillar matrix, the TDT muscles show it to a certain extent, and other adult muscles to a greater extent, particularly the leg muscles. These findings rely exclusively on focused ion beam SEM, and a modest concern is that it might be valuable to also observe these events using more traditional EM approaches, such as TEM of longitudinal sections. Such data would add experimental robustness to the very impressive data that are already presented.

We thank the reviewer for this suggestion of validating our FIB-SEM results by another imaging method. In addition to the sarcomere branching in the NCDN KD IFMs already shown by confocal microscopy in the original manuscript, we have now provided additional confocal microscopy images showing sarcomere branching in NCDN overexpression (OE) muscles. Additionally, we have now included longitudinal 2D EM examples of sarcomere branching in Supplementary Figure 1 which should facilitate examination of previously published TEM data of *Drosophila* muscles for examples of sarcomere branching.

In the second part of the manuscript, the authors determine if knockdown of candidate genes that are known or thought to modulate muscle fiber identity also correspondingly transform the myofibrillar matrix. Indeed knockdown of *salm* in the flight muscles, which was previously shown to result in a transformation of IF muscles to tubular muscles, also results in the appearance of a matrix in the transformed IF muscles. Similarly, knockdown of H15, not previously shown to regulate muscle fiber fate, also causes acquisition of leg a muscle-like network in the TDT. Finally, knockdown of Neurochondrin causes the appearance of the myofibrillar matrix in the IF muscles.

These studies are significant in showing the existence of the myofibrillar matrix in insect muscles, its diverse occurrence across different muscle types, and some aspects of its genetic control. However the genetic approach was not as thorough: while the knockdown data were compelling, there were no corresponding gain of function approaches, such as determining if over-expression of neurochondrin, H15 or *salm* would have effects opposite to their loss of function phenotypes.

We agree that overexpression data is critical to a better understanding of how each gene regulates myofibrillar connectivity. *Salm* OE was previously shown using light microscopy (Schonbauer et al,

Nature, 2011) to convert the tubular leg muscles towards a more fibrillar phenotype, and we have recently repeated these results in Katti et al. (bioRxiv, 2021). Since the Salm KD FIB-SEM data from Figure 3 was collected with the aim of demonstrating that we would get the expected result when converting a fibrillar to a tubular muscle, we did not feel that performing the opposite experiment by FIB-SEM would add anything significant to the already available light microscopy data regarding our understanding of Salm specification of muscle fate.

Regarding H15, we recently showed that H15 OE alone had no apparent effect on the contractile or mitochondrial networks in the flight, jump, or leg muscles as assessed by standard confocal microscopy (Katti et al. bioRxiv, 2021 doi: <https://doi.org/10.1101/2021.09.30.462204>). However, H15 OE was able to rescue the Salm KD phenotype in the flight muscles and return them to fibrillar muscles with no branching in 38% of the muscle fibers assessed (Data from Katti et al. included below). Thus, H15 appears able to play a role in myofibrillar connectivity in both directions. We have now added some discussion relating the above mentioned H15 and Salm OE data to the myofibrillar network data presented here.

m

n

r

At the time of our original submission, we were unable to find any neurochondrin OE lines from any of the stock centers or in the literature. However, thanks to the reviewers suggestion, we have now made our own NCDN OE line. Briefly, UAS-NCDN (Neurochondrin) transgenic flies were generated using the plasmid containing BDGP Tagged NCDN ORF (UFO10226; obtained from DGRC). We confirmed the fragment size by performing digestion with EcoRI and further amplified the fragment by PCR and verified by sequencing. The UAS-NCDN transgenic flies were generated using phiC31-mediated integration into the attP2 landing site, and the injection of embryos was performed by Bestgene Inc (USA). Crossing to a muscle-specific (Mef2-Gal4) line resulting in a relatively small (compared to NCDN KD), but significant overexpression of NCDN (57% by qPCR). Despite the relatively small increase in NCDN expression, NCDN OE indirect flight muscles also show a fraying phenotype similar to the NCDN KD flight muscles.

The fraying and merging of adjacent myofibrils as noted by the authors in the NCDN knockdowns are indeed interesting and not unique. In addition to the recent papers the authors cite, it is notable that haploinsufficiency for several myofibrillar protein genes can also result in fraying and merging of adjacent myofibrils (see for example Beall et al 1989, Genes Dev). Since the authors do not investigate the cellular mechanisms that control branching, is it possible that the NCDN knockdown simply dysregulates actin or myosin assembly?

We thank the reviewer for pointing out this missed citation and for this suggestion. We have now added the citation to the discussion of fraying. Additionally, we have now performed qPCR to assess the expression level of 11 transcripts related to actin and myosin assembly in the NCDN KD muscles. These data reveal that while there is no apparent change in myosin heavy chain expression, IFM specific actin isoform (Act88F) expression is reduced in the NCDN KD muscles compared to wild type IFMs. Additionally, using the weeP26 MHC-GFP line (Orfanos and Sparrow. J Cell Sci, 2013), we find that the IFM myosin variant is not converted by NCDN KD the way it is for Salm KD. Finally, we also show that the NCDN KD mediated loss of Act88F does not lead to a compensatory upregulation of the tubular isoform, Act79B. Thus, NCDN KD appears to lead to an actin-myosin assembly imbalance which, as shown in Beall et al., is sufficient to induce sarcomere branching in IFMs as pointed out by the reviewer. The induction of fraying by NCDN OE is also consistent with NCDN serving to regulate balance during myofibril assembly.

Overall this is an interesting study that enables the analysis of a novel but significant phenomenon in a genetically tractable organism. However, there are weaknesses with the work in its current state due to a lack of genetic rigor and a failure to provide insight into the underlying molecular control of myofibrillar matrix formation.

We thank the reviewer for this constructive criticism. We have now improved our genetic rigor by including discussion of our recent H15 OE data included in Katti et al. (BioRxiv, 2021) and the previous Salm OE data from Frank Schnorrer and in Katti et al, as well as creating a new NCDN OE line. Additionally, thanks to the reviewers suggestions, we have now included mechanistic data in the new Figure 5 which suggests that NCDN regulates the balance between actin and myosin expression in flight muscles as well as demonstrated that the regulatory role of NCDN for myofibrillar connectivity occurs between 24 and 48 hours APF when myofibril assembly takes place.

Additional comments/questions.

1. Did the authors focus upon a specific direct flight muscle for their analyses, and if so which muscle was this?

We thank the reviewer for this question. While we did not specifically track which DFM muscles were used, based on the small size (only five cells per muscle) and the thin, elongated shape of the muscle, the muscle we used appears most consistent with DFM50 (from Jahrling et al. Front Sys Neuro, 2010). For reference, we have included below a 3D reconstruction of DFM muscles from Jahrling et al. as well as our own SEM images of the DFM muscle we used prior to and after FIB-SEM imaging.

[redacted]

2. Line 196: should “towards” be “away from”?

We thank the reviewer for catching this error in the text which we have now corrected. *“Overall, these data suggest that H15 regulates the connectivity of myofibrillar networks in the tubular TDT muscle by preventing these muscles from taking a leg muscle phenotype.”*

3. Figures 2, 3, and 4 lack comparable control images, since these images are at a different magnification from those shown in Figure 1.

We thank the reviewer for this suggestion and have now included wild type images at similar magnifications in Figures 2-4.

4. Figures 2-4, please could that authors draw boxes around each graph as it is difficult to see where one graph ends and the next one starts.

We thank the reviewer for pointing this out. We have now added lines separating each graph within the figures as well as moved some panels from Figure 4 to the new Supplemental Figure 4 to aid in the clarity of the figures.

5. Line 261-262, I do not see any evidence that would support the description of H15 as a “fibrillar muscle specification” factor.

We thank the reviewer for catching this error. We have now included the citation to Katti et al. bioRxiv, 2021 which demonstrates that loss of H15 in IFM causes a fibrillar to tubular switch (figure also included below) and that H15 OE can rescue the fibrillar to tubular switch caused by salm KD (figure already included above).

6. Figure 4, panels c and d. These should be at the same magnification.

These panels are now at the same magnification.

Reviewer #3 (Remarks to the Author):

Glancy and colleagues describe the branching pattern of myofibrils in different *Drosophila* muscles by FIB-SEM. They have previously done similar work in vertebrate muscles and now show that branching is conserved in insect muscles (TDT and leg).

Overall, the imaging and the data analysis is very solidly done. Nevertheless, one does not learn much from this paper.

We thank the reviewer for these comments. We hope that readers will learn that myofibrillar networks are conserved in *Drosophila* muscles and that the connectivity of these networks can be regulated by cell type dependent (Salm & H15) and independent (NCDN) mechanisms as information on myofibrillar connectivity in this genetically tractable organism was not previously available in the literature. Additionally, we have now provided new mechanistic insight into how NCDN regulates sarcomere branching by altering the actin-myosin balance within the muscle as well as that NCDN acts during the myofibril assembly stage of development. In addition to the overall data on *Drosophila* myofibrillar networks, this paper provides the first information on the specific role of NCDN in muscle function in any species.

H15 and the adjacent gene midline encode closely related and recently duplicated T-box transcription factors that are co-regulated and have similar functions in *Drosophila*. Why is this not mentioned anywhere? Does a midline KD have a similar phenotype?

We thank the reviewer for this question and agree that midline should be mentioned in relation to H15 in our work. Whereas H15 KD flies were completely flightless (Katti et al. BioRxiv, 2021 doi: <https://doi.org/10.1101/2021.09.30.462204>), Mid KD flies were capable of flight albeit weaker than in wild type flies. Moreover, while H15 KD causes a fibrillar to tubular switch in IFMs, and a parallel to grid-like mitochondrial network conversion in the jump and leg Fiber 1 muscles, respectively (Katti et al, BioRxiv, 2021), Mid KD does not causes conversion of myofibrillar or mitochondrial networks in these muscles. As the comparable H15 KD data for this new Mid KD data is already part of Katti et al, BioRxiv, 2021, we have provided the data for the reviewer below and will include it as a supplemental figure in Katti et al. which we are also currently revising for Nature Communications (this approach has been okayed by Dr. Evan Bardot, the editor of both manuscripts).

The *salm* phenotype is not surprising, given that *salm* is upstream of *H15* and is the master regulator of IFM fate.

We agree that the *salm* phenotype is not surprising as the goal of the *salm* studies here was to confirm that the system worked as expected based on the results in Figure 1, i.e. if we converted a fibrillar muscle to tubular, the myofibrils would become connected. Because there was no prior information that the tubular muscles formed highly connected myofibrillar networks, it was critical to establish that, in addition to wild type tubular muscles, muscles which normally would be fibrillar also form myofibrillar networks when converted to a tubular type. Thus, the *salm* data serve as a positive control showing that we can turn on/off myofibrillar networks by genetically converting between

fibrillar and tubular contractile types.

The neurochondrin phenotype is exciting, but somewhat haphazardly introduced and not necessarily convincing. I find it curious how the authors measure cross-sectional area. To me the most apparent phenotype of neurochondrin KD appears to be myofibrils with a bigger diameter (bigger cross-sectional area), which is quite apparent from 4b and d. Only by counting every “wisp” of actomyosin separating from a myofibril as a myofibril do they get the smaller diameter. The phenotype they describe is correct, but perhaps the interpretation could also be that the main phenotype of neurochondrin KD is an increase of myofibril cross-sectional area, which eventually leads to instability and fraying of myofibrils. In summary, this phenotype looks different from the branching documented in Figure 1 (branches are much thinner and more frequent), making me doubt that it is the same phenotype.

We thank the reviewer for these comments. The cross-sectional data was originally assessed per myofibril segment which, as the reviewer points out, skews the data towards the many small segments rather than the fewer much larger segments. However, in order to work around this issue, we performed a volume weighted analysis which provides greater weight to the larger objects. The volume weighted analyses show that the mean myofibril cross-sectional area is larger in the NCDN KD flight muscles as noted by the reviewer. We have now marked the volume-weighted average in the figure which highlights the larger myofibril size in the NCDN KD flight muscles.

Also, we agree with the reviewer that the specifics of how branching occurs is different in the NCDN KD IFMs compared to the rest of the muscles in the paper. This is because all of the other branching muscles are tubular in nature whereas the NCDN KD IFMs retain their fibrillar nature based on the mostly circular myofibril shape, presence of fibrillar molecular markers (Salm and H15), preserved mitochondrial organization (many large mitochondria in parallel to contractile axis), and the presence of tracheoles within the muscle fiber. Also, we now show that NCDN KD does not cause a fibrillar to tubular conversion of the actin isoform or myosin heavy chain variant in IFMs. However, the phenotype of interest in this paper is the formation of myofibrillar networks linked together by sarcomere branches which are clearly shown in Figure 4 and the supporting supplemental figures and videos.

Based on the comments from this reviewer as well as reviewer #2, we now show that NCDN KD leads to reduced actin expression (both Act88F and Act79B) without altering myosin heavy chain expression resulting in an unbalancing of the actin/myosin ratio which was previously shown to cause fraying myofibrils. Moreover, we also find that this actin-myosin unbalancing and the resultant branching phenotype only occurs when NCDN is reduced during the myofibril assembly developmental stage.

A more descriptive paper without the neurochondrin phenotype might be more valuable for the community, in simply documenting the fact of myofibril branching in some muscle types in *Drosophila*.

We hope the addition of mechanistic insights into the role of neurochondrin in regulating the actin/myosin balance within flight muscles and the identification of the developmental timepoints in which neurochondrin acts to regulate sarcomere branching are sufficient to convince the reviewer that the neurochondrin phenotype is robust and worth including within this paper.

REVIEWERS' COMMENTS

Reviewer #1 (Remarks to the Author):

The authors have addressed all of my concerns. I would like to thank the authors for both addressing my comments and providing answers to my questions. I look forward to seeing this paper published.

Reviewer #2 (Remarks to the Author):

The authors have done an effective job of addressing my concerns. I have only one remaining comment, that I think the authors can address in the text. In the experiments where the authors vary the timing of NCDN knockdown, they determine that a relatively early (but not late) knockdown triggers the formation of the thin sarcomere branches. This seems surprising since at this earlier stage the nascent myofibrils are quite small, and are generally thought to grow from the center of the myofibril outwards. As a result, how can an early knockdown cause a phenotype in the regions of the sarcomere that have not formed yet (i.e., the periphery of the mature sarcomeres)? The authors might discuss this and whether the timing of onset of Gal4 activity correlates effectively with the timing of loss of NCDN protein or transcripts.

Reviewer #3 (Remarks to the Author):

The authors have addressed my technical concerns, especially with respect to the neurochondrin phenotype. The demonstration that the neurochondrin phenotype is owing to an imbalance of actin and myosin again indicates it is unrelated to the phenotypic differences between tubular and fibrillar muscles.

REVIEWERS' COMMENTS

Reviewer #1 (Remarks to the Author):

The authors have addressed all of my concerns. I would like to thank the authors for both addressing my comments and providing answers to my questions. I look forward to seeing this paper published.

Reviewer #2 (Remarks to the Author):

The authors have done an effective job of addressing my concerns. I have only one remaining comment, that I think the authors can address in the text. In the experiments where the authors vary the timing of NCDN knockdown, they determine that a relatively early (but not late) knockdown triggers the formation of the thin sarcomere branches. This seems surprising since at this earlier stage the nascent myofibrils are quite small, and are generally thought to grow from the center of the myofibril outwards. As a result, how can an early knockdown cause a phenotype in the regions of the sarcomere that have not formed yet (i.e., the periphery of the mature sarcomeres)? The authors might discuss this and whether the timing of onset of Gal4 activity correlates effectively with the timing of loss of NCDN protein or transcripts.

We thank the reviewer for this comment. The different stage specific NCDN knockdowns (Figure 5a-e) were all imaged at adulthood after myofibril assembly and maturation take place, and this has now been clarified in the text for the Act88F-Gal4 driven knockdown (it was already mentioned for the other Gal4s), which is the phenotype discussed here. Act88F-Gal4 is expected to begin knockdown at ~24 hours APF and continue through adulthood and thus should encompass the entirety of the myofibril assembly process as described by the reviewer. Thus, the lack of NCDN throughout myofibril assembly leads to the formation of thin branches in the adult flight muscle. Consistently, the myofibrillar phenotype when knocking down NCDN starting at the beginning of myogenesis (Mef2-Gal4) begins at ~32 hours APF which is during the myofibril assembly stage (Figure 5f-u). Additionally, by taking advantage of the transcriptional resource developed by Spletter et al. (Elife, 2018), we can now show that NCDN transcript expression (part of Cluster 34 shown below) peaks sharply at ~30 hours APF. This has now been added to the results section. Overall, three separate measures each suggest that NCDN is primarily operating from 24-48 hours APF, or during myofibril assembly.

[redacted]

Reviewer #3 (Remarks to the Author):

The authors have addressed my technical concerns, especially with respect to the neurochondrin phenotype. The demonstration that the neurochondrin phenotype is owing to an imbalance of actin and

myosin again indicates it is unrelated to the phenotypic differences between tubular and fibrillar muscles.